# Experience transforms crossmodal object representations in the anterior temporal lobes

Aedan Yue Li[1]*, Natalia Ladyka-Wojcik[1], Heba Qazilbash[1], Ali Golestani[2], Dirk B Walther[1,3], Chris B Martin[4], Morgan D Barense[1,3]

[1]Department of Psychology, University of Toronto, Toronto, Canada; [2]Department of Physics and Astronomy, University of Calgary, Calgary, Canada; [3]Rotman Research Institute, Baycrest Health Sciences, North York, Canada; [4]Department of Psychology, Florida State University, Tallahassee, United States

**Abstract** Combining information from multiple senses is essential to object recognition, core to the ability to learn concepts, make new inferences, and generalize across distinct entities. Yet how the mind combines sensory input into coherent crossmodal representations – the *crossmodal binding problem* – remains poorly understood. Here, we applied multi-echo fMRI across a 4-day paradigm, in which participants learned three-dimensional crossmodal representations created from well-characterized unimodal visual shape and sound features. Our novel paradigm decoupled the learned crossmodal object representations from their baseline unimodal shapes and sounds, thus allowing us to track the emergence of crossmodal object representations as they were learned by healthy adults. Critically, we found that two anterior temporal lobe structures – temporal pole and perirhinal cortex – differentiated learned from non-learned crossmodal objects, even when controlling for the unimodal features that composed those objects. These results provide evidence for integrated crossmodal object representations in the anterior temporal lobes that were different from the representations for the unimodal features. Furthermore, we found that perirhinal cortex representations were by default biased toward visual shape, but this initial visual bias was attenuated by crossmodal learning. Thus, crossmodal learning transformed perirhinal representations such that they were no longer predominantly grounded in the visual modality, which may be a mechanism by which object concepts gain their abstraction.

*For correspondence:
aedanyue.li@utoronto.ca

## Editor's evaluation

The fMRI study is important because it investigates fundamental questions about the neural basis of multimodal binding using an innovative multi-day learning approach. The results provide solid evidence for learning-related changes in the anterior temporal lobe. This paper is of interest to a broad audience of cognitive neuroscientists.

## Introduction

The world is a great blooming, buzzing confusion (*James, 1890*) of the senses. Our ability to understand 'what is out there' depends on combining sensory features to form *crossmodal object concepts*. A child, for example, might form the concept 'frog' by learning that the visual appearance of a four-legged creature goes with the sound of its croaking. Consequently, this child has also learned that frogs do not produce barking sounds, as the child has created a unique object association for a frog from specific unimodal shape and sound features. Forming coherent crossmodal object representations is

thus essential for human experience, allowing adaptive behavior under changing environments. Yet, how is it possible for the child to know that the sound of croaking is associated with the visual shape of a frog, even when she might be looking at a dog? How does the human mind form meaningful concepts from the vast amount of unimodal feature information that bombards the senses, allowing us to interpret our external world?

Known as the *crossmodal binding problem*, this unresolved question in the cognitive sciences concerns how the mind combines unimodal sensory features into coherent crossmodal object representations. Better characterization of how this computational challenge is solved will not only improve our understanding of the human mind but will also have important consequences for the design of future artificial neural networks. Current artificial machines do not yet reach human performance on tasks involving crossmodal integration (*Guo et al., 2019*; *Fei et al., 2022*) or generalization beyond previous experience, (*Keysers et al., 2020*; *Hupkes et al., 2020*; *Santoro et al., 2017*) which are limitations thought to be in part driven by the inability of existing machines to resolve the binding problem (*Greff et al., 2020*).

One theoretical view from the cognitive sciences suggests that crossmodal objects are built from component unimodal features represented across distributed sensory regions. (*Barsalou, 2008*) Under this view, when a child thinks about 'frog', the visual cortex represents the appearance of the shape of the frog, whereas the auditory cortex represents the croaking sound. Alternatively, other theoretical views predict that multisensory objects are not only built from their component unimodal sensory features, but that there is also a crossmodal integrative code that is different from the sum of these parts (*Patterson et al., 2007*; *Saksida and Bussey, 2010*; *Cowell et al., 2019*; *Kent et al., 2016*; *Damasio, 1989*). These latter views propose that anterior temporal lobe structures can act as

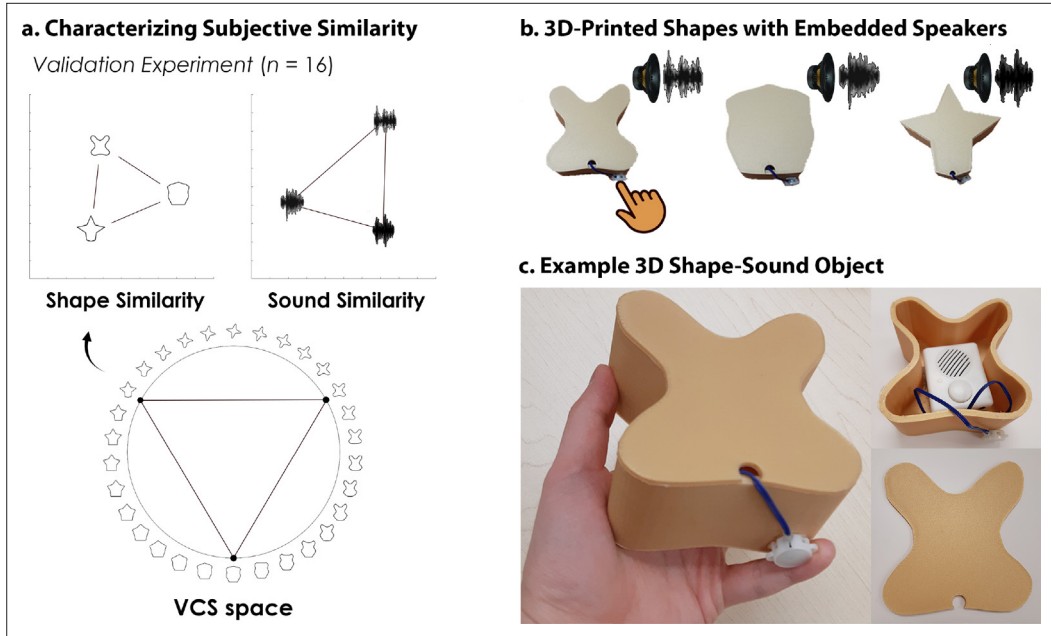

**Figure 1.** 3D-printed objects. An independent validation experiment ensured that the similarity of the selected shapes and sounds were well-matched. (**a**) Three shapes were sampled from the *Validated Circular Shape (VCS) Space* (shown as black points on VCS space), (*Li et al., 2020*) a stimulus space whereby angular distance corresponds to subjective shape similarity. Three sounds were sampled from a set of five experimenter-created sounds. This independent validation experiment ensured that we could characterize the change in similarity structure following crossmodal learning, because we knew the baseline similarity structure that is, two triangular representational geometries visualized using multidimensional scaling (*Shepard, 1980*; also see *Figure 2— figure supplement 1*). Furthermore, this procedure ensured that the subjective similarity of the three features was equated within each modality. (**b**) The shapes were then 3D-printed with a hollow space and embedded with a button-activated speaker. (**c**) Participants could physically explore and palpate the 3D shape-sound objects. Critically, we manipulated whether the button-activated speaker was operational across learning days (see Methods/*Figure 2*).

a polymodal 'hub' that combines separate features into integrated wholes (*Patterson et al., 2007*; *Cowell et al., 2019*; *Suzuki and Naya, 2014*; *Ralph et al., 2017*).

Thus, a key theoretical challenge central to resolving the crossmodal binding problem is understanding how anterior temporal lobe structures form object representations. Are crossmodal objects entirely built from features distributed across sensory regions, or is there also integrative coding in the anterior temporal lobes? Furthermore, the existing literature has predominantly studied the neural representation of well-established object concepts from the visual domain alone (*Barsalou, 2008*; *Patterson et al., 2007*; *Saksida and Bussey, 2010*; *Cowell et al., 2019*; *Kent et al., 2016*; *Damasio, 1989*; *Suzuki and Naya, 2014*; *Ralph et al., 2017*; *Ferko et al., 2022*; *Bausch et al., 2021*; *Pagan et al., 2013*; *Yee and Thompson-Schill, 2016*; *Barense et al., 2012*; *Erez et al., 2016*; *Liang et al., 2020*; *Martin et al., 2018*; *Hebart et al., 2020*; *Li et al., 2022*), even though human experience is fundamentally crossmodal.

Here, we leveraged multi-echo fMRI (*Kundu et al., 2017*) across a novel 4-day task in which participants learned to associate unimodal visual shape and sound features into 3D crossmodal object representations. First, we characterized shape (*Li et al., 2020*) and sound features in a separate validation experiment, ensuring that the unimodal features were well-matched in terms of their subjective similarity (*Figure 1*). On the learning task, participants independently explored the 3D-printed shapes and heard novel experimenter-constructed sounds. The participants then learned specific shape-sound associations (congruent objects), while other shape-sound associations were not learned (incongruent objects).

Critically, our 4-day learning task allowed us to isolate neural activity associated with integrative coding in anterior temporal lobe structures that emerges with experience and differs from the neural patterns recorded at baseline. The learned and non-learned crossmodal objects were constructed from the same set of three validated shape and sound features, ensuring that factors such as familiarity with the unimodal features, subjective similarity, and feature identity were tightly controlled (*Figure 2*). If the mind represented crossmodal objects entirely as the reactivation of unimodal shapes and sounds (i.e. objects are constructed from their parts) then there should be no difference between the learned and non-learned objects (because they were created from the same three shapes and sounds). By contrast, if the mind represented crossmodal objects as something over and above their component features (i.e. representations for crossmodal objects rely on integrative coding that is different from the sum of their parts) then there should be behavioral and neural differences between learned and non-learned crossmodal objects (because the only difference across the objects is the learned relationship between the parts). Furthermore, this design allowed us to determine the relationship between the object representation acquired *after* crossmodal learning and the unimodal feature representations acquired *before* crossmodal learning. That is, we could examine whether learning led to abstraction of the object representations such that it no longer resembled the unimodal feature representations.

In brief, we found that crossmodal object concepts were represented as distributed sensory-specific unimodal features along the visual and auditory processing pathways, as well as integrative crossmodal combinations of those unimodal features in the anterior temporal lobes. Intriguingly, the perirhinal cortex – an anterior temporal lobe structure – was biased toward the visual modality before crossmodal learning at baseline, with greater activity toward shape over sound features. Pattern similarity analyses revealed that the shape representations in perirhinal cortex were initially unaffected by sound, providing evidence of a default visual shape bias. However, crossmodal learning transformed the object representation in perirhinal cortex such that it was no longer predominantly visual. These results are consistent with the idea that the object representation had become abstracted away from the component unimodal features with learning, such that perirhinal representations was no longer grounded in the visual modality.

## Results
### Four-day crossmodal object learning task
Measuring within-subject changes after crossmodal learning
We designed a 4-day learning task where each participant learned a set of shape-sound associations that created crossmodal objects (*Figure 2*). There were two days involving only behavioral measures (*Day 1* and *Day 3*). Before crossmodal learning on Day 1, participants explored the 3D-printed shapes

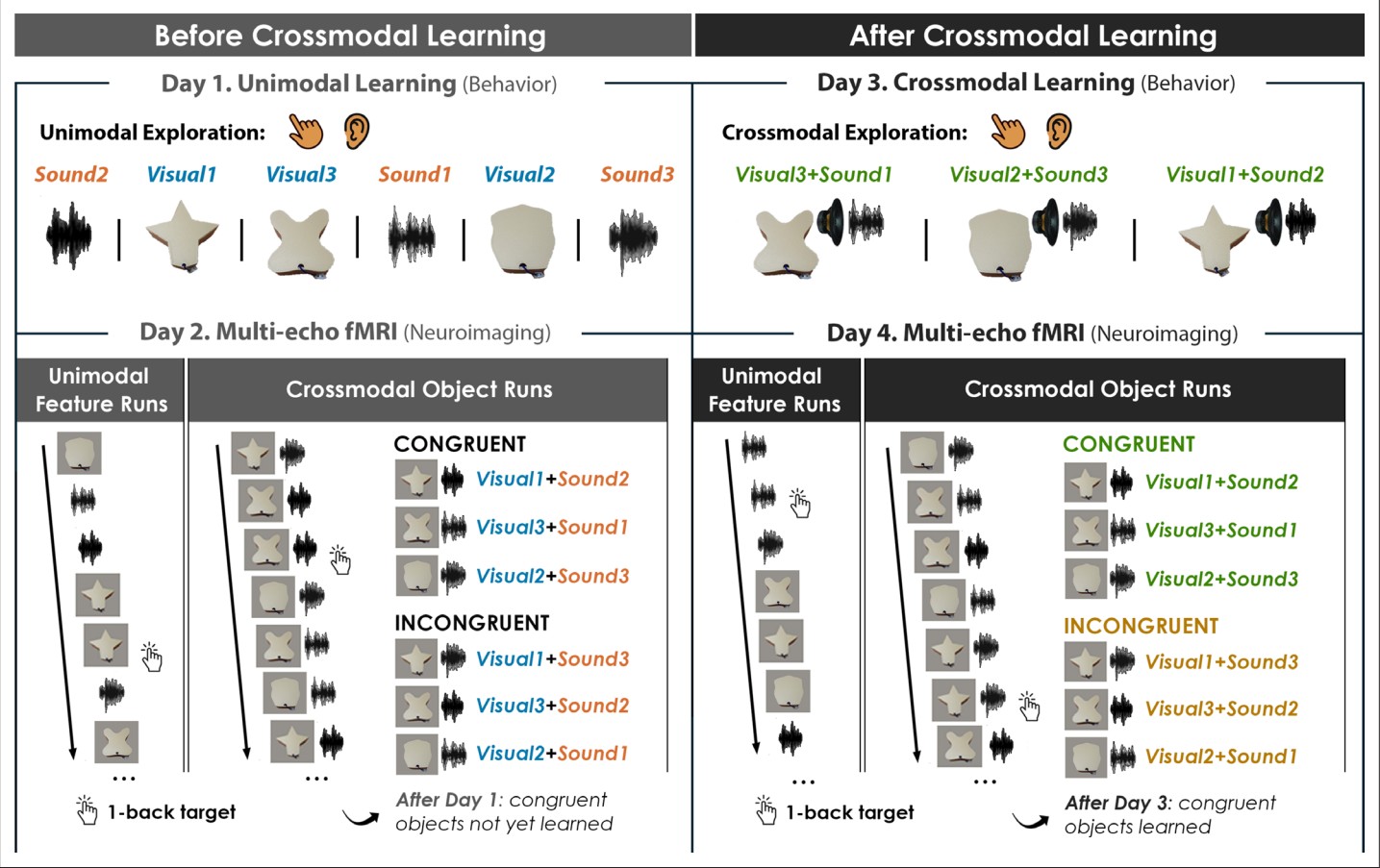

**Figure 2.** Four-day crossmodal object learning task. On Day 1 (behavior), participants heard sounds through a headset and explored 3D-printed shapes while the button-activated speakers were not operational. During a separate task (*Figure 2—figure supplement 1*), participants rated the similarity of the visual shapes and sound features. On Day 2 (neuroimaging), participants completed (**i**) 10 Unimodal Feature runs in which they performed a 1-back task involving the shape and sound features experienced separately and (ii) 5 Crossmodal Object runs in which they performed a 1-back task for the shapes and sounds experienced simultaneously. As participants at this point have not yet learned the congruent shape-sound pairings, the Day 2 neuroimaging session serves as a within-subject neural baseline for how the unimodal features were represented before crossmodal learning. On Day 3 (behavior), participants again explored the shape and sound features. Participants now learned to make crossmodal associations between the specific visual and sound features that composed the shape-sound object by pressing the button to play an embedded speaker, thus forming congruent object representations (i.e. crossmodal learning). Shape-sound associations were counterbalanced across participants, and we again collected similarity ratings between the shapes and sounds on a separate task. On Day 4 (neuroimaging), participants completed the same task as on Day 2. In summary, across 4 days, we characterized the neural and behavioral changes that occurred before and after shapes and sounds were paired together to form crossmodal object representations. As the baseline similarity structure of the shape and sound features were a priori defined (see *Figure 1*) and measured on the first day of learning (see *Figure 2—figure supplement 1*), changes to the within-subject similarity structure provide insight into whether the crossmodal object representations (acquired after crossmodal learning) differed from component unimodal representations (acquired before crossmodal learning).

The online version of this article includes the following figure supplement(s) for figure 2:

**Figure supplement 1.** Pairwise similarity task and results.

(*Visual*) and heard the sounds (*Sound*) separately. In blocks of trials interleaved with these exploration phases, participants rated the similarity of the shapes and sounds (see *Figure 2—figure supplement 1*). During crossmodal learning on Day 3, participants explored specific shape-sound associations (*Congruent* objects) by pressing the button on each 3D-printed shape to play the associated sound, with pairings counterbalanced across observers. Again, the participants rated the similarity of the shapes and sounds. Notably, all participants could recognize their specific shape-sound associations at the end of Day 3, confirming that the congruent shape-sound objects were successfully learned (performance = 100% for all participants).

 There were two neuroimaging days (*Day 2* and *Day 4*), during which we recorded brain responses to unimodal features presented separately and to unimodal features presented simultaneously using

multi-echo fMRI (*Figure 2*). During Unimodal Feature runs, participants either viewed images of the 3D-printed shapes or heard sounds. During Crossmodal Object runs, participants experienced either the shape-sound associations learned on Day 3 (*Congruent*) or shape-sound associations that had not been learned on Day 3 (*Incongruent*). We were especially interested in neural differences between congruent and incongruent objects as evidence of crossmodal integration; experience with the unimodal features composing congruent and incongruent objects was equated and the only way to distinguish them was in terms of how the features were integrated.

## Behavioral pattern similarity
### Subjective similarity changes after crossmodal learning
To understand how crossmodal learning impacts behavior, we analyzed the within-subject change in subjective similarity of the unimodal features *before* (Day 1) and *after* (Day 3) participants learned their crossmodal pairings (*Figure 2*). In other words, we determined whether the perceived similarity of the unimodal feature representations changed after participants had experienced those unimodal features combined into crossmodal objects.

We conducted a linear mixed model which included learning day (before vs. after crossmodal learning) and congruency (congruent vs. incongruent) as fixed effects. We observed a robust learning-related behavioral change in terms of how participants experienced the similarity of shape and sound features (*Figure 2—figure supplement 1*): there was a main effect of learning day (before or after crossmodal learning: $F_{1,51} = 24.45$, p<0.001, $\eta^2=0.32$), a main effect of congruency (congruent or incongruent: $F_{1,51} = 6.93$, p=0.011, $\eta^2=0.12$), and an interaction between learning day and congruency ($F_{1,51} = 15.33$, p<0.001, $\eta^2=0.23$). Before crossmodal learning, there was no difference in similarity between congruent and incongruent shape-sound features ($t_{17}=0.78$, p=0.44), whereas after crossmodal learning, participants rated shapes and sounds associated with congruent objects to be more similar than shapes and sounds associated with incongruent objects ($t_{17}=5.10$, p<0.001, *Cohen's d*=1.28; *Figure 2—figure supplement 1*). Notably, this learning-related change in similarity was observed in 17 out of 18 participants. We confirmed this experience-dependent change in similarity structure in a separate behavioral experiment with a larger sample size (observed in 38 out of 44 participants; learning day x congruency interaction: $F_{1,129} = 13.74$, p<0.001; $\eta^2=0.096$; *Figure 2—figure supplement 1*).

## Whole-brain univariate analysis
### Unimodal shape and sound representations are distributed
In the first set of neuroimaging analyses**,** we examined whether distributed brain regions were involved in representing unimodal shapes and sounds. During unimodal runs (shapes and sounds presented separately), we observed robust bilateral modality-specific activity across the neocortex (*Figure 3a–c*). The ventral visual stream extending into the perirhinal cortex activated more strongly to unimodal visual compared to sound information, indicating that perirhinal cortex activity was by default biased toward visual information in the unimodal runs (i.e. toward complex visual shape configurations; *Figure 3a*). The auditory processing stream, from the primary auditory cortex extending into the temporal pole along the superior temporal sulcus, activated more strongly to unimodal sound compared to visual information (*Figure 3b*). These results replicate the known representational divisions across the neocortex and show that regions processing unimodal shapes and sounds are distributed across visual and auditory processing pathways.*Barense et al., 2016*; *Mishkin et al., 1983*; *Poremba and Mishkin, 2007*. Furthermore, the robust signal quality we observe in anterior temporal regions demonstrates the improved quality of the multi-echo ICA pipeline employed in the current study, as these anterior temporal regions are often susceptible to signal dropout with standard single echo designs due to magnetic susceptibility issues near the sinus air/tissue boundaries (*Figure 3—figure supplement 1*).

## Region-of-interest univariate analysis
### Anterior temporal lobes differentiate between congruent and incongruent conditions
We next examined univariate activity focusing on five a priori regions thought to be important for representing unimodal features and their integration (*Patterson et al., 2007*; *Cowell et al., 2019*)

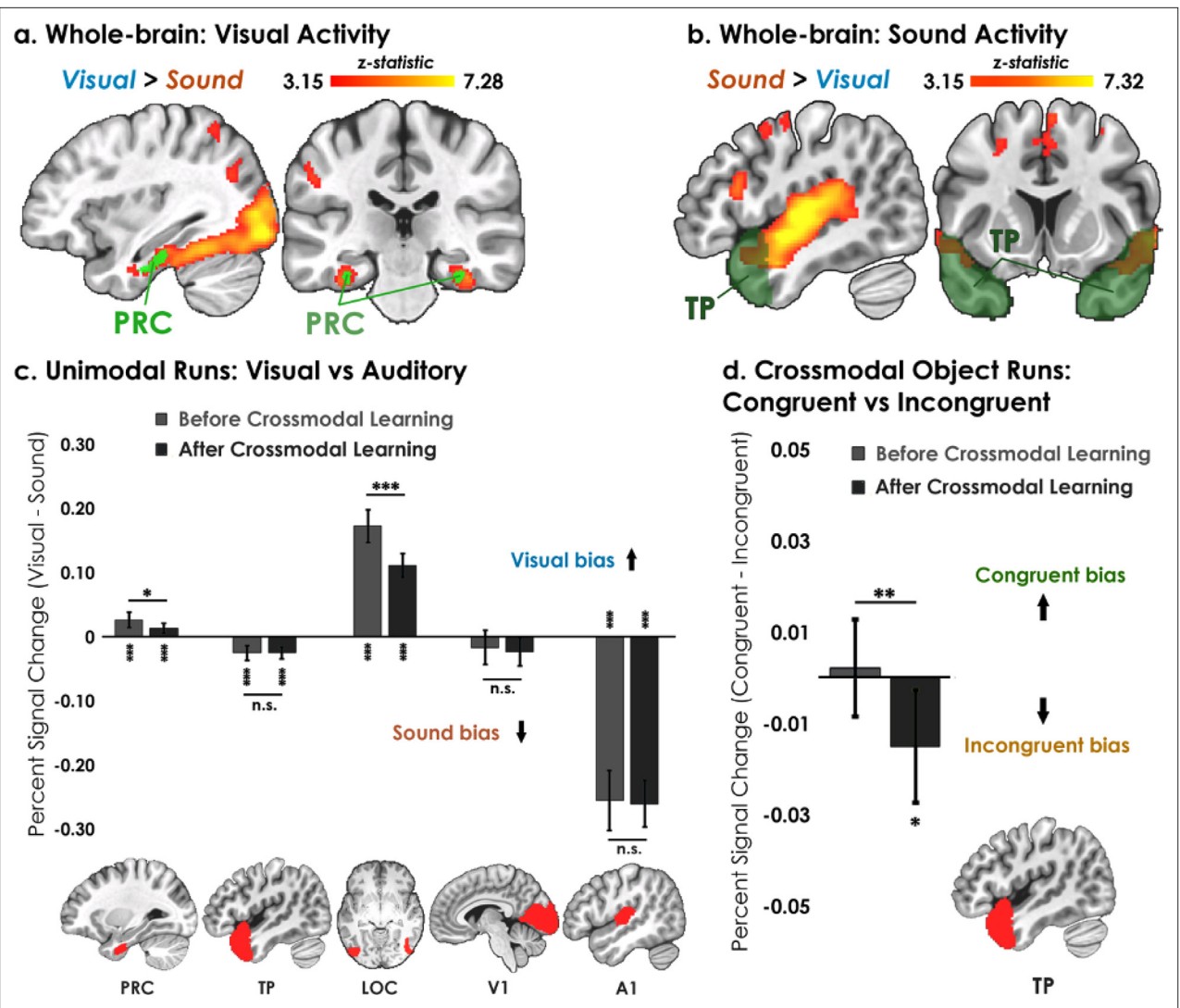

**Figure 3.** Univariate results. (**a–b**) Univariate analyses superimposed on MNI-152 standard space. All contrasts were thresholded at voxel-wise p=0.001 and cluster-corrected at p=0.05 (random-effects, FSL FLAME; 6 mm spatial smoothing). Collapsing across learning days, robust modality-specific activity was observed across the neocortex. (**c–d**) Five ROIs were a priori selected based on existing theory: *Patterson et al., 2007*; *Cowell et al., 2019* temporal pole – TP, perirhinal cortex – PRC, lateral occipital complex – LOC, primary visual cortex – V1, and primary auditory cortex – A1. (**c**) Consistent with the whole-brain results, LOC was biased toward visual features whereas A1 and TP were biased toward sound features. Activation in PRC and LOC showed learning-related shifts, with the magnitude of visual bias decreasing after crossmodal learning. (**d**) TP was the only brain region to show an experience-dependent change in univariate activity to the learned shape-sound associations during crossmodal object runs. * p<0.05, ** p<0.01, *** p<0.001. Asterisks above or below bars indicate a significant difference from zero. Horizontal lines within brain regions reflect an interaction between modality or congruency with learning day (e.g. reduction in visual bias after crossmodal learning in PRC). Error bars reflect the 95% confidence interval (n = 17).

The online version of this article includes the following figure supplement(s) for figure 3:

**Figure supplement 1.** Signal quality comparison from a representative participant.

temporal pole, perirhinal cortex, lateral occipital complex (LOC), primary visual cortex (V1), and primary auditory cortex (A1). For each ROI, we conducted a linear mixed model which included learning day (before vs. after crossmodal learning) and modality (visual vs. sound feature) as fixed factors. Collapsing across learning days, perirhinal cortex ($t_{67}$=5.53, p<0.001, *Cohen's d*=0.67) and LOC ($t_{63}$=16.02, p<0.001, *Cohen's d*=2.00) were biased toward visual information, whereas temporal pole ($t_{67}$=6.73, p<0.001, *Cohen's d*=0.82) and A1 ($t_{67}$=17.09, p<0.001, *Cohen's d*=2.07) were biased toward sound information (*Figure 3d*). Interestingly, we found a small overall bias toward sound in V1, consistent with past work (*Vetter et al., 2014*; $t_{67}$=2.26, p=0.027, *Cohen's d*=0.20). Next, we

determined how neural responses in these regions changed following crossmodal learning. We observed an interaction between learning day and modality in perirhinal cortex ($F_{1,48}$ = 5.24, p=0.027, $\eta^2$=0.098) and LOC ($F_{1,45}$ = 25.89, p<0.001, $\eta^2$=0.37; **Figure 3d**). These regions activated more strongly to visual information at baseline before crossmodal learning compared to after crossmodal learning, indicative of a visual bias that was attenuated with experience.

As a central goal of our study was to identify brain regions that were influenced by the learned crossmodal associations, we next examined univariate differences between *Congruent vs. Incongruent* for crossmodal object runs as a function of whether the crossmodal association had been learned. We conducted a linear mixed model for each ROI which included learning day (before vs. after crossmodal learning) and congruency (congruent vs. incongruent objects) as fixed factors. We observed a significant interaction between learning day and congruency in the temporal pole ($F_{1,48}$ = 7.63, p=0.0081, $\eta^2$=0.14). Critically, there was no difference in activity between congruent and incongruent objects at baseline before crossmodal learning ($t_{33}$=0.37, p=0.72), but there was more activation to incongruent compared to congruent objects after crossmodal learning ($t_{33}$=2.42, p=0.021, *Cohen's d*=0.42). As the unimodal shape-sound *features* experienced by participants were the same before and after crossmodal learning (**Figure 2**), this finding reveals that the univariate signal in the temporal pole was differentiated between congruent and incongruent objects that had been constructed from the same unimodal features.

By contrast, we did not observe a univariate difference between the congruent and incongruent conditions in the perirhinal cortex, LOC, V1, or A1 ($F_{1,45-48}$ between 0.088 and 2.34, *p* between 0.13 and 0.77). Similarly, the exploratory ROIs hippocampus (HPC: $F_{1,48}$ = 0.32, p=0.58) and inferior parietal lobe (IPL: $F_{1,48}$ = 0.094, p=0.76) did not distinguish between the congruent and incongruent conditions.

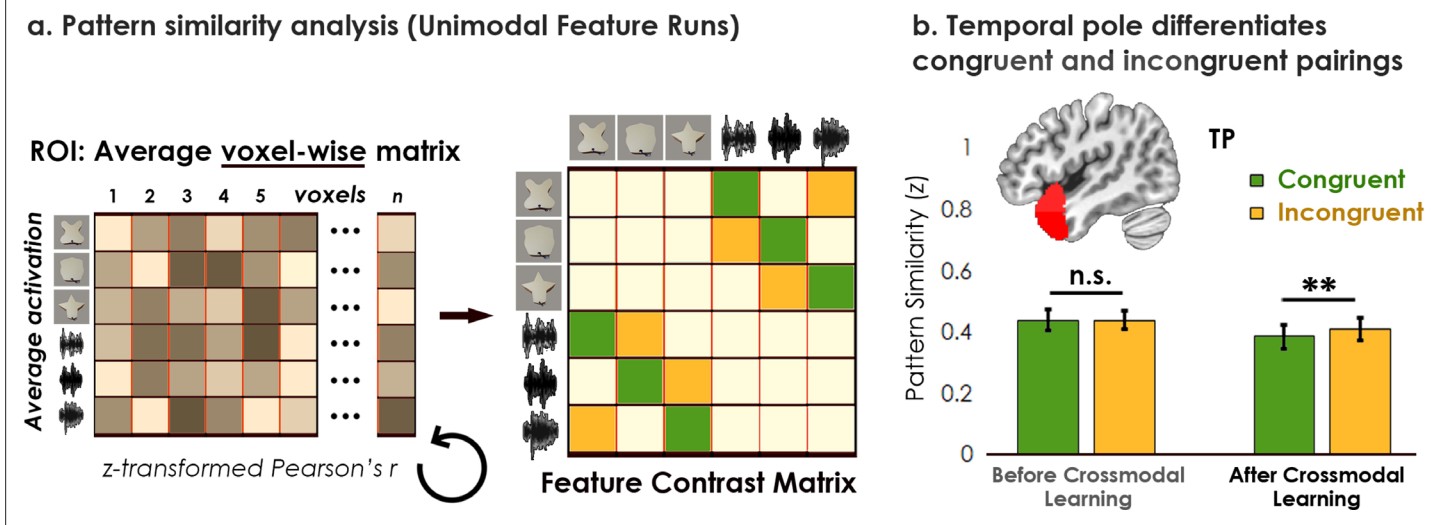

**Figure 4.** Pattern similarity analyses for unimodal feature runs. (**a**) Contrast matrix comparing the effect of congruency on feature representations. The voxel-wise matrix averaged across unimodal runs were autocorrelated using the z-transformed Pearson's correlation, creating a unimodal feature-level contrast matrix. We examined the average pattern similarity between unimodal features associated with congruent objects (green) compared to the same unimodal features associated with incongruent objects (yellow). (**b**) Pattern similarity analysis revealed an interaction between learning day and congruency in the temporal pole (TP). At baseline before crossmodal learning, there was no difference in neural similarity between unimodal features that paired to create congruent objects compared to the same unimodal features that paired to create incongruent objects. After crossmodal learning, however, there was *less* neural similarity between the unimodal features of pairs comprising congruent objects compared to the unimodal features of pairs comprising incongruent objects. Because congruent and incongruent objects were built from the same shapes and sounds, this result provides evidence that learning about crossmodal object associations influenced the representations of the component features in the temporal pole. There was no difference between the congruent and incongruent pairings in any other ROI (**Figure 4—figure supplement 1**). ** p<0.01. Error bars reflect the 95% confidence interval (n = 17).

The online version of this article includes the following figure supplement(s) for figure 4:

**Figure supplement 1.** Pattern similarity analyses between unimodal features associated with congruent objects and incongruent objects, before and after crossmodal learning (analysis visualized in **Figure 4** in the main text).

## Neural pattern similarity

### Congruent associations differ from incongruent associations in anterior temporal lobes

We next conducted a series of representational similarity analyses across Unimodal Feature and Crossmodal Object runs before and after crossmodal learning. Here, we investigated whether representations for unimodal features were changed after learning the crossmodal associations between those features (i.e. learning the crossmodal pairings that comprised the shape-sound objects). Such a finding could be taken as evidence that learning crossmodal *object* concepts transforms the original representation of the component unimodal *features*. More specifically, we compared the correlation between congruent and incongruent shape-sound features within Unimodal Feature runs before and after crossmodal learning (*Figure 4a*).

We conducted a linear mixed model which included learning day (before vs. after crossmodal learning) and congruency (congruent vs. incongruent) as fixed effects for each ROI. Complementing the previous behavioral pattern similarity results (*Figure 2—figure supplement 1*), in the temporal pole we observed a main effect of learning day (before or after crossmodal learning: $F_{1,32} = 4.63$, p=0.039, $\eta^2=0.13$), a main effect of congruency (congruent or incongruent object: $F_{1,64} = 7.60$, p=0.0076, $\eta^2=0.11$), and an interaction between learning day and congruency ($F_{1,64} = 6.09$, p=0.016, $\eta^2=0.087$). At baseline before crossmodal learning, there was no difference in pattern similarity between congruent features compared to incongruent features in the temporal pole ($t_{33}=0.22$, p=0.82). After crossmodal learning, however, there was lower pattern similarity for shape and sound features associated with congruent compared to incongruent objects ($t_{33}=3.47$, p=0.0015, *Cohen's d*=0.22; *Figure 4*). Thus, although in behavior we observed that learning the crossmodal associations led to greater pattern similarity between congruent compared to incongruent features (*Figure 2—figure supplement 1*), this *greater behavioral similarity* was related to *reduced neural similarity* following crossmodal learning in the temporal pole.

By contrast, the other four a priori determined ROIs (perirhinal cortex, LOC, V1, or A1) did not show an interaction between learning day and congruency ($F_{1,60-64}$ between 0.039 and 1.30, *p* between 0.26 and 0.84; *Figure 4—figure supplement 1*). Likewise, our two exploratory ROIs (hippocampus, inferior parietal lobe) did not show an interaction between learning day and congruency ($F_{1,64}$ between 0.68 and 0.91, *p* between 0.34 and 0.41; *Figure 5—figure supplement 1*).

## The visually biased code in perirhinal cortex was attenuated with learning

The previous analyses found that the temporal pole differentiated between congruent and incongruent shape-sound pairs after participants learned the crossmodal pairings. Next, we characterized how the representations of these unimodal features changed after they had been paired with features from another stimulus modality to form the crossmodal objects. Our key question was whether learning crossmodal associations transformed the unimodal feature representations.

First, the voxel-wise activity for unimodal feature runs was correlated to the voxel-wise activity for crossmodal object runs at baseline before crossmodal learning (*Figure 5a*). Specifically, we quantified the similarity in the patterns for the visual *shape features* with the *crossmodal objects* that had that same shape, as well as between the *sound features* and the *crossmodal objects* that had that same sound. We then conducted a linear mixed model which included modality (visual vs. sound) as a fixed factor within each ROI. Consistent with the univariate results (*Figure 3*), we observed greater pattern similarity when there was a match between sound features in the temporal pole ($F_{1,32} = 15.80$, p<0.001, $\eta^2=0.33$) and A1 ($F_{1,32} = 145.73$, p<0.001, $\eta^2=0.82$), and greater pattern similarity when there was a match in the visual shape features in the perirhinal cortex ($F_{1,32} = 10.99$, p=0.0023, $\eta^2=0.26$), LOC ($F_{1,30} = 20.09$, p<0.001, $\eta^2=0.40$), and V1 ($F_{1,32} = 22.02$, p<0.001, $\eta^2=0.41$). Pattern similarity for each ROI was higher for one of the two modalities, indicative of a baseline modality-specific bias toward either visual or sound content.

We then examined whether the original representations would change after participants learned how the features were paired together to make specific crossmodal objects, conducting the same analysis described above after crossmodal learning had taken place (*Figure 5b*). With this analysis, we sought to measure the relationship between the representation for the learned crossmodal object and the original baseline representation for the unimodal features. More specifically, the voxel-wise

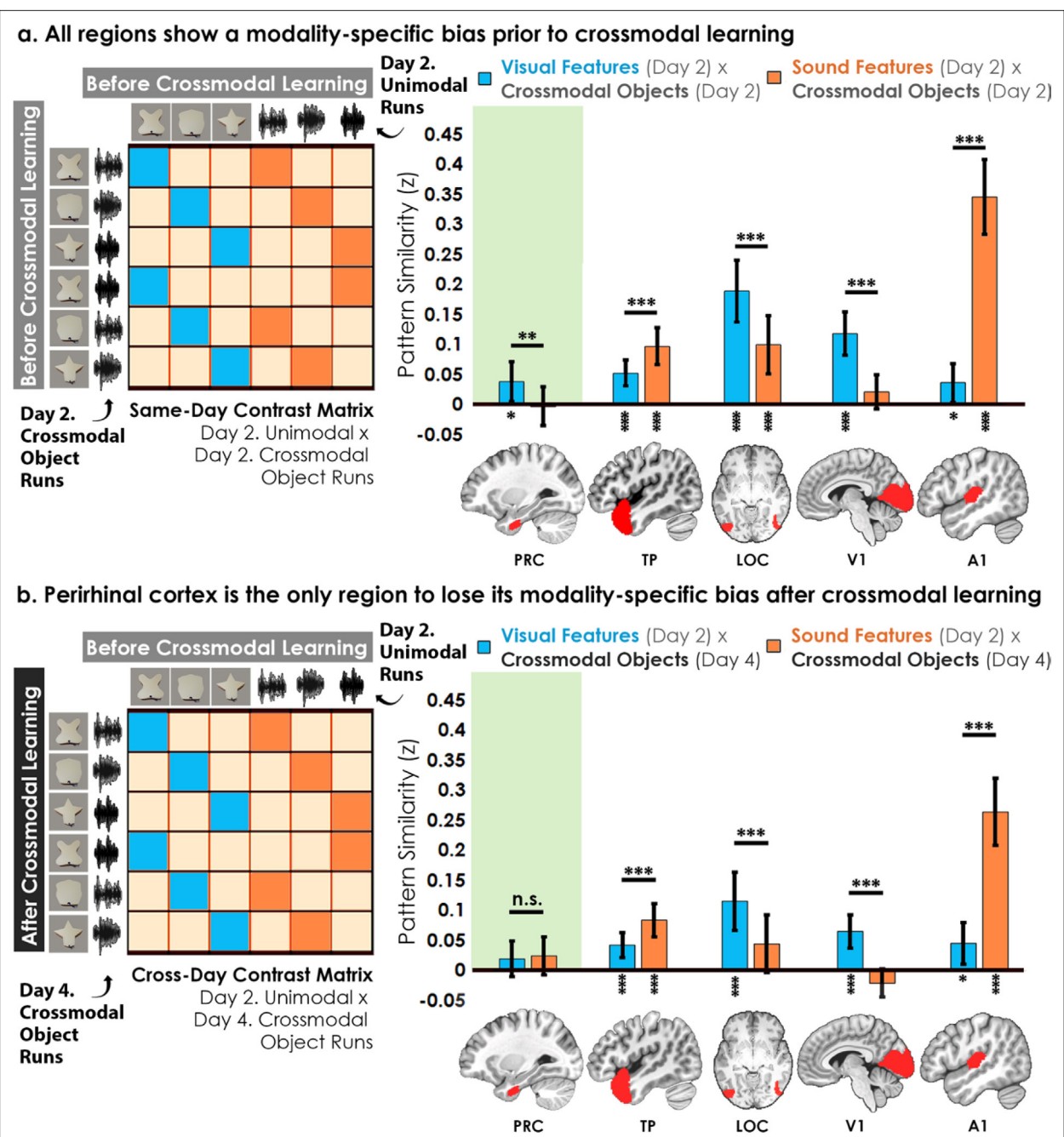

**Figure 5.** Contrast matrices and pattern similarity analyses investigating the effect of crossmodal learning on modality-specific biases. The voxel-wise matrix for unimodal feature runs on Day 2 were correlated to the voxel-wise matrix for crossmodal object runs on (**a**) Day 2 and (**b**) Day 4, creating a contrast matrix between visual and auditory unimodal features to crossmodal objects that contained those features. We compared the average pattern similarity (z-transformed Pearson correlation) between shape (blue) and sound (orange) features across learning days. (**a**) Robust modality-specific feature biases were observed in all examined regions before crossmodal learning. That is, pattern similarity for each brain region was higher for one of the two modalities, indicative of a modality-specific bias. For example, pattern similarity in perirhinal cortex (PRC) preferentially tracked the visual features of the crossmodal objects, evidence of a default visual shape bias *before crossmodal learning*. (**b**) Critically, we found that perirhinal representations were transformed with experience, such that the initial visual bias was attenuated *after crossmodal learning* (i.e. denoted by a significant interaction, shown by shaded green regions), evidence that representations were no longer predominantly grounded in the visual modality. * $p < 0.05$, ** $p < 0.01$, *** $p < 0.001$. Horizontal lines within brain regions indicate a significant main effect of modality. Vertical asterisks denote pattern similarity comparisons relative to 0. Error bars reflect the 95% confidence interval (n = 17).

The online version of this article includes the following figure supplement(s) for figure 5:

**Figure supplement 1.** Analyses for the hippocampus (HPC) and inferior parietal lobe (IPL).

*Figure 5 continued on next page*

*Figure 5 continued*

**Figure supplement 2.** The voxel-wise matrix for Unimodal Feature runs on Day 4 were correlated to the voxel-wise matrix for Crossmodal Object runs on Day 4 (see *Figure 5* in the main text for an example).

activity for unimodal feature runs *before* crossmodal learning was correlated to the voxel-wise activity for crossmodal object runs *after* crossmodal learning (*Figure 5b*). Another linear mixed model which included modality as a fixed factor within each ROI revealed that the perirhinal cortex was no longer biased toward visual shape after crossmodal learning ($F_{1,32}$ = 0.12, p=0.73), whereas the temporal pole, LOC, V1, and A1 remained biased toward either visual shape or sound ($F_{1,30-32}$ between 16.20 and 73.42, all p<0.001, $\eta$ *Guo et al., 2019* between 0.35 and 0.70).

To investigate this effect in perirhinal cortex more specifically, we conducted a linear mixed model to directly compare the change in the visual bias of perirhinal representations from before crossmodal learning to after crossmodal learning (green regions in *Figure 5a vs. 5b*). Specifically, the linear mixed model included learning day (before vs. after crossmodal learning) and modality (visual feature match to crossmodal object vs. sound feature match to crossmodal object). Results revealed a significant interaction between learning day and modality in the perirhinal cortex ($F_{1,775}$ = 5.56, p=0.019, $\eta^2$=0.071), meaning that the baseline visual shape bias observed in perirhinal cortex (green region of *Figure 5a*) was significantly attenuated with experience (green region of *Figure 5b*). After crossmodal learning, a given shape no longer invoked significant pattern similarity between objects that had the same shape but differed in terms of what they sounded like. Taken together, these results suggest that prior to learning the crossmodal objects, the perirhinal cortex had a default bias toward representing the visual shape information and was not representing sound information of the crossmodal objects. After crossmodal learning, however, the visual shape bias in perirhinal cortex was no longer present. That is, with crossmodal learning, the representations within perirhinal cortex started to look less like the visual features that comprised the crossmodal objects, providing evidence that the perirhinal representations were no longer predominantly grounded in the visual modality.

To examine whether these results differed by congruency (i.e. whether any modality-specific biases differed as a function of whether the object was congruent or incongruent), we conducted exploratory linear mixed models for each of the five a priori ROIs across learning days. More specifically, we correlated: (1) the voxel-wise activity for Unimodal Feature Runs *before* crossmodal learning to the voxel-wise activity for Crossmodal Object Runs *before* crossmodal learning (Day 2 vs. Day 2), (2) the voxel-wise activity for Unimodal Feature Runs *before* crossmodal learning to the voxel-wise activity for Crossmodal Object Runs *after* crossmodal learning (Day 2 vs Day 4), and (3) the voxel-wise activity for Unimodal Feature Runs *after* crossmodal learning to the voxel-wise activity for Crossmodal Object Runs *after* crossmodal learning (Day 4 vs Day 4). For each of the three analyses described, we then conducted separate linear mixed models which included modality (visual feature match to crossmodal object vs. sound feature match to crossmodal object) and congruency (congruent vs. incongruent).

There was no significant relationship between modality and congruency in any ROI between Day 2 and Day 2 ($F_{1,346-368}$ between 0.00 and 1.06, *p* between 0.30 and 0.99), between Day 2 and Day 4 ($F_{1,346-368}$ between 0.021 and 0.91, *p* between 0.34 and 0.89), or between Day 4 and Day 4 ($F_{1,346-368}$ between 0.01 and 3.05, *p* between 0.082 and 0.93). However, exploratory analyses revealed that perirhinal cortex was the only region without a modality-specific bias and where the unimodal feature runs were not significantly correlated to the crossmodal object runs *after crossmodal learning* (*Figure 5—figure supplement 2*).

Taken together, the overall pattern of results suggests that representations of the crossmodal objects in perirhinal cortex were heavily influenced by their consistent visual features *before* crossmodal learning. However, the crossmodal object representations were no longer influenced by the component visual features *after* crossmodal learning (*Figure 5*, *Figure 5—figure supplement 2*). Additional exploratory analyses did not find evidence of experience-dependent changes in the hippocampus or inferior parietal lobes (*Figure 5—figure supplement 1*).

Importantly, the change in pattern similarity in the perirhinal cortex across learning days (*Figure 5*) is unlikely to be driven by noise, poor alignment of patterns across sessions, or generally reduced responses. Other regions with numerically similar pattern similarity to perirhinal cortex did not change across learning days (e.g. visual features x crossmodal objects in A1 in *Figure 5*; the exploratory ROI

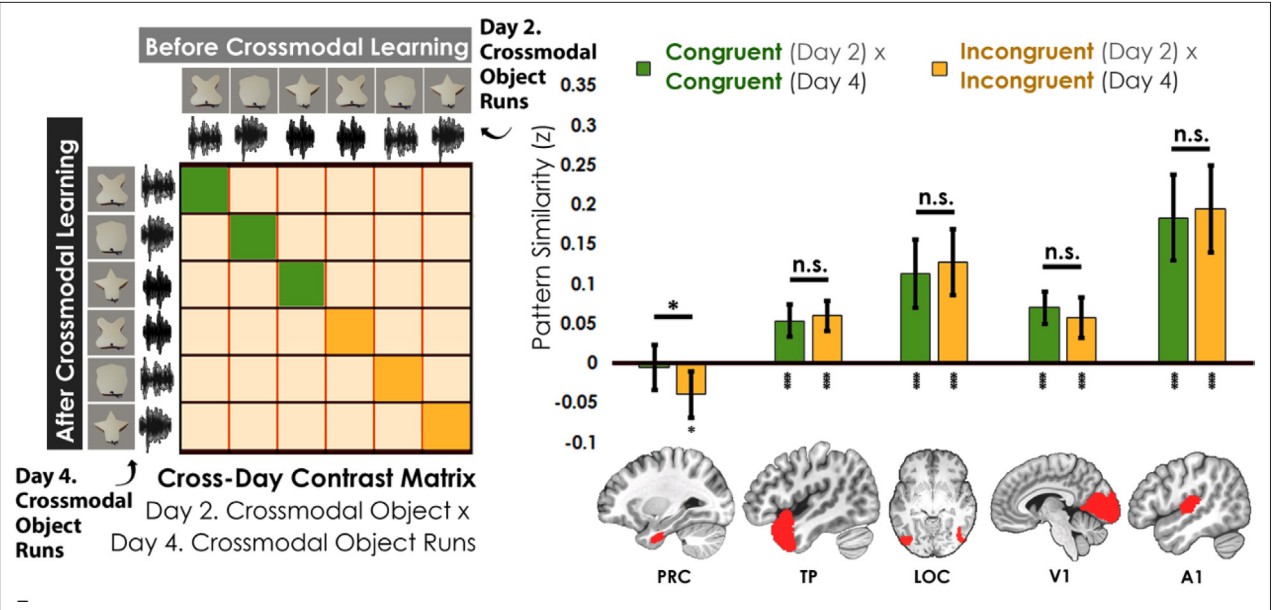

**Figure 6.** Contrast matrix shown on the left panel, with actual results shown on the right panel. We compared the average pattern similarity across learning days between crossmodal object runs on Day 2 with crossmodal object runs on Day 4 (z-transformed Pearson correlation). We observed lower average pattern similarity for incongruent objects (yellow) compared to congruent (green) objects in perirhinal cortex (PRC). These results suggest that perirhinal cortex differentiated congruent and incongruent objects constructed from the same features. Furthermore, pattern similarity was never above 0 for the perirhinal cortex. By contrast, there was no significant difference between congruent and incongruent objects in any other examined region, and pattern similarity was always above 0. * denotes p<0.05, ** denotes p<0.01, *** denotes p<0.001. Horizontal lines within brain regions denote a main effect of congruency. Vertical asterisks denote pattern similarity comparisons relative to 0. Error bars reflect the 95% confidence interval (n = 17).

hippocampus with numerically similar pattern similarity to perirhinal cortex also did not change in *Figure 5—figure supplement 1*).

## Representations in perirhinal cortex change with experience

So far, we have shown that the perirhinal cortex was by default biased toward visual shape features (*Figure 5a*), and that this visual shape bias was attenuated with experience (*Figure 5b*; *Figure 5—figure supplement 2*). In the final analysis, we tracked how the *individual crossmodal object representations* themselves change after crossmodal learning.

We assessed the cross-day pattern similarity between Crossmodal Object Runs by correlating the congruent and incongruent runs across learning days (*Figure 6*). We then conducted a linear mixed model which included congruency (congruent vs. incongruent) as a fixed factor for each a priori ROI. Perirhinal cortex was the only region that differentiated between congruent and incongruent objects in this analysis (PRC: $F_{1,34}$ = 4.67, p=0.038, $\eta^2$=0.12; TP, LOC, V1, A1: $F_{1,32-34}$ between 0.67 and 2.83, *p* between 0.10 and 0.42). Pattern similarity in perirhinal cortex did not differ from 0 for congruent objects across learning days ($t_{35}$=0.39, p=0.70) but was significantly lower than 0 for incongruent objects ($t_{35}$=2.63, p=0.013, *Cohen's d*=0.44). By contrast, pattern similarity in temporal pole, LOC, V1, and A1 was significantly correlated across learning days (pattern similarity >0; $t_{33-35}$ between 4.31 and 6.92 all p<0.001) and did not differ between congruent and incongruent objects (temporal pole, LOC, V1, and A1; $F_{1,32-34}$ between 0.67 and 2.83, *p* between 0.10 and 0.42). Thus, perirhinal cortex was unique in that it not only differentiated between congruent and incongruent objects that were built from the same unimodal features (i.e. representations of the whole crossmodal object that was different than the unimodal features that composed it), but it also showed no significant pattern similarity above 0 for the same representations across learning days (i.e. suggesting that the object representations were transformed after crossmodal learning).

No significant difference between the congruent and incongruent conditions were observed for the hippocampus ($F_{1,34}$ = 0.34, p=0.56) or inferior parietal lobe ($F_{1,34}$ = 0.00, p=0.96) in a follow-up exploratory analysis (*Figure 5—figure supplement 1*).

## Discussion

Known as the *crossmodal binding problem*, a long-standing question in the cognitive sciences has asked how the mind forms coherent concepts from multiple sensory modalities. To study this problem, we designed a 4-day task to decouple the learned crossmodal object representations (Day 3 and 4) from the baseline unimodal shape and sound features (Day 1 and 2). We equated the familiarity, subjective similarity, and identity of the unimodal feature representations composing the learned (congruent) and unlearned (incongruent) objects, ensuring that any differences between the two would not be driven by single features but rather by the integration of those features (*Figure 2*). Paired with multi-echo fMRI to improve signal quality in the anterior temporal lobes (*Figure 3—figure supplement 1*), this novel paradigm tracked the emergence of crossmodal object concepts from component baseline unimodal features in healthy adults.

We found that the temporal pole and perirhinal cortex – two anterior temporal lobe structures – came to represent new crossmodal object concepts with learning, such that the acquired crossmodal object representations were different from the representation of the constituent unimodal features (*Figures 5 and 6*). Intriguingly, the perirhinal cortex was by default biased toward visual shape, but that this initial visual bias was attenuated with experience (*Figures 3c and 5*, *Figure 5—figure supplement 2*). Within the perirhinal cortex, the acquired crossmodal object concepts (measured after crossmodal learning) became less similar to their original component unimodal features (measured at baseline before crossmodal learning); *Figures 5 and 6*, *Figure 5—figure supplement 2*. This is consistent with the idea that object representations in perirhinal cortex integrate the component sensory features into a whole that is different from the sum of the component parts, which might be a mechanism by which object concepts obtain their abstraction.

As one solution to the crossmodal binding problem, we suggest that the temporal pole and perirhinal cortex form unique crossmodal object representations that are different from the distributed features in sensory cortex (*Figures 4–6*, *Figure 5—figure supplement 2*). However, the nature by which the integrative code is structured and formed in the temporal pole and perirhinal cortex following crossmodal experience – such as through transformations, warping, or other factors – is an open question and an important area for future investigation. Furthermore, these distinct anterior temporal lobe structures may be involved with integrative coding in different ways. For example, the crossmodal object representations measured after learning were found to be related to the component unimodal feature representations measured before learning in the temporal pole but not the perirhinal cortex (*Figures 5 and 6*, *Figure 5—figure supplement 2*). Moreover, pattern similarity for congruent shape-sound pairs were lower than the pattern similarity for incongruent shape-sound pairs after crossmodal learning in the temporal pole but not the perirhinal cortex (*Figure 4b*, *Figure 4—figure supplement 1*). As one interpretation of this pattern of results, the temporal pole may represent new crossmodal objects by combining previously learned knowledge (*Barsalou, 2008*; *Patterson et al., 2007*; *Saksida and Bussey, 2010*; *Cowell et al., 2019*; *Damasio, 1989*; *Suzuki and Naya, 2014*; *Ralph et al., 2017*; *Binder and Desai, 2011*). Specifically, research into *conceptual combination* has linked the anterior temporal lobes to compound object concepts such as 'hummingbird' (*Lynott and Connell, 2010*; *Coutanche et al., 2020*; *Baron and Osherson, 2011*). For example, participants during our task may have represented the sound-based 'humming' concept and visually based 'bird' concept on Day 1, forming the crossmodal 'hummingbird' concept on Day 3; *Figures 1 and 2*, which may recruit less activity in temporal pole than an incongruent pairing such as 'barking-frog'. For these reasons, the temporal pole may form a crossmodal object code based on pre-existing knowledge, resulting in reduced neural activity (*Figure 3d*) and pattern similarity toward features associated with learned objects (*Figure 4b*).

By contrast, perirhinal cortex may be involved in pattern separation following crossmodal experience. In our task, participants had to differentiate congruent and incongruent objects constructed from the same three shape and sound features (*Figure 2*). An efficient way to solve this task would be to form distinct object-level outputs from the overlapping unimodal feature-level inputs such that congruent objects are made to be orthogonal from the representations before learning (i.e. measured as pattern similarity equal to 0 in the perirhinal cortex; *Figures 5b and 6*, *Figure 5—figure supplement 2*), whereas non-learned incongruent objects could be made to be dissimilar from the representations before learning (i.e. anticorrelation, measured as patten similarity less than 0 in the perirhinal cortex; *Figure 6*). Because our paradigm could decouple neural responses to the learned object

representations (on Day 4) from the original component unimodal features at baseline (on Day 2), these results could be taken as evidence of pattern separation in the human perirhinal cortex (*Cowell et al., 2019*; *Kent et al., 2016*). However, our pattern of results could also be explained by other types of crossmodal integrative coding. For example, incongruent object representations may be less stable than congruent object representations, such that incongruent objects representation are warped to a greater extent than congruent objects (*Figure 6*).

Our results suggest that the temporal pole and perirhinal cortex are involved in representing crossmodal objects after a period of crossmodal learning. Although this observation is consistent with previous animal research (*Jacklin et al., 2016*) finding that a period of experience is necessary for the perirhinal cortex to represent crossmodal objects, future work will need to determine whether our findings are driven by *only* experience or by experience *combined with* sleep-dependent consolidation (*Schapiro et al., 2017*). Perhaps, a future study could explore how separate unimodal features and the integrative object representations change over the course of the same learning day compared to multiple learning days after sleep. Nevertheless, perirhinal cortex was critically influenced by experience, potentially explaining why findings in this literature have been at times mixed, as stimulus history was not always controlled across different experiments (*Taylor et al., 2006*; *Holdstock et al., 2009*). In our study, we explicitly controlled for stimulus history (*Figure 2*), ensuring that participants extensively explored individual features by the end of the first day and formed crossmodal objects by the end of the third day.

Complementing seminal patient work causally linking anterior temporal lobe damage to the loss of object concepts (*Hodges and Patterson, 1997*), we show that the formation of new crossmodal concepts also recruits anterior temporal lobe structures like the temporal pole and perirhinal cortex. An important direction of future work will be to investigate the fine-grained functional divisions within the heterogeneous anterior temporal lobe region. One recent study has found that the anterior temporal lobe can be separated into 34 distinct functional regions (*Persichetti et al., 2021*), suggesting that a simple temporal pole versus perirhinal cortex division may not fully capture the complexity of this region. Imaging the anterior temporal lobe has long been known to be challenging with functional neuroimaging due to signal dropout (*Visser et al., 2010*). We show that a multi-echo fMRI sequence (*Kundu et al., 2017*) may be especially useful in future work, as multi-echo fMRI mitigates signal dropout better than the standard single-echo fMRI (see *Figure 3—figure supplement 1* for a visual comparison).

Importantly, the initial visual shape bias observed in the perirhinal cortex was attenuated by experience (*Figure 5*, *Figure 5—figure supplement 2*), suggesting that the perirhinal representations had become abstracted and were no longer predominantly grounded in a single modality after crossmodal learning. One possibility may be that the perirhinal cortex is by default visually driven as an extension to the ventral visual stream, (*Saksida and Bussey, 2010*; *Cowell et al., 2019*; *Kent et al., 2016*) but can act as a polymodal 'hub' region for additional crossmodal input following learning. A complementary possibility may be that our visual features contained tactile information (*Figure 1c*) that the perirhinal cortex may be sensitive to following the initial exploration phase on our task (*Figure 2*; *Holdstock et al., 2009*) Critically, other brain regions like the LOC also reduced in visual bias (*Figure 3c*), which may reflect visual imagery or feedback connectivity between the anterior temporal lobes. However, the perirhinal cortex was the only region where the visual bias was entirely attenuated following crossmodal learning (*Figure 5b*).

An interesting future line of investigation may be to explore whether there exist similar changes to the visual bias in artificial neural networks that aim to learn crossmodal object concepts (*Guo et al., 2019*; *Fei et al., 2022*; *Greff et al., 2020*). Previous human neuroimaging has shown that the anterior temporal lobes are important for intra-object configural representations, (*Yeung et al., 2017*; *Watson and Lee, 2013*) such that damage to the perirhinal cortex (*Barense et al., 2012*; *Bonnen et al., 2021*) leads to object discrimination impairment. For example, human participants with perirhinal cortex damage are unable to resolve feature-level interference created by viewing multiple objects with overlapping features. Certain types of errors made by deep learning models (*Guo et al., 2022*) also seem to resemble the kinds of errors made by human patients, (*Barense et al., 2012*; *Taylor et al., 2006*; *Hodges and Patterson, 1997*; *Bonnen et al., 2021*) whereby accurate object recognition can be disrupted by feature-level interference. Writing the word 'iPod' on an apple image, for instance, can lead to deep learning models falsely recognizing the apple as an actual iPod (*Goh et al.,*

*2021*). As certain limitations of existing neural networks may be driven by an inability to resolve the binding problem (*Greff et al., 2020*), future work to mimic the coding properties of anterior temporal lobe structures may allow artificial machines to better mimic the remarkable human ability to learn concepts, make new inferences, and generalize across distinct entities.

Notably, our perirhinal cortex mask overlaps with a key region of the ventral anterior temporal lobe thought to be the central locus of crossmodal integration in the 'hub and spokes' model of semantic representations (*Patterson et al., 2007Ralph et al., 2017*). However, additional work has also linked other brain regions to the convergence of unimodal representations, such as the hippocampus (*Butler and James, 2011*; *Clouter et al., 2017*; *Viganò and Piazza, 2020*) and inferior parietal lobes (*Viganò et al., 2021*; *Binder and Desai, 2011*). This past work on the hippocampus and inferior parietal lobe does not necessarily address the crossmodal binding problem that was the main focus of our present study, as previous findings often do not differentiate between crossmodal integrative coding and the convergence of unimodal feature representations per se. Furthermore, previous studies in the literature typically do not control for stimulus-based factors such as experience with unimodal features, subjective similarity, or feature identity that may complicate the interpretation of results when determining regions important for crossmodal integration. Indeed, we found evidence consistent with the convergence of unimodal feature-based representations in both the hippocampus and inferior parietal lobes (*Figure 5—figure supplement 1*), but no evidence of crossmodal integrative coding different from the unimodal features. The hippocampus and inferior parietal lobes were both sensitive to visual and sound features before and after crossmodal learning (see *Figure 5—figure supplement 1*). Yet, the hippocampus and inferior parietal lobes did not differentiate between the congruent and incongruent conditions or change with experience (see *Figure 5—figure supplement 1*).

In summary, forming crossmodal object concepts relies on the representations for the whole crossmodal object in anterior temporal lobe structures different from the distributed unimodal feature representations in sensory regions. It is this hierarchical architecture that supports our ability to understand the external world, providing one solution to the age-old question of how crossmodal concepts can be constructed from their component features.

## Methods

The experiments described in this study were approved by the University of Toronto Ethics Review Board (protocols 37590 and 38856). Informed consent was obtained for all participants in the study prior to their participation.

### Initial stimulus validation experiment

#### Participants
16 participants (Females = 11, $M_{age}$ = 18.63 years) were recruited from the University of Toronto undergraduate participant pool and from the community. Course credit or $10 /hr CAD was provided as compensation.

#### Stimuli
Three shape stimuli were sampled from the Validated Shape Space (*Li et al., 2020*) at equidistant positions, ensuring that the shapes were equated in their subjective similarity. The sound stimuli were manually generated in a similar procedure to how the shape stimuli from the Validated Shape Space (*Li et al., 2020*) were originally created. More specifically, distinct sounds were morphed together to create 5 complex, unrecognizable sounds that lasted for a duration of 2 s.

#### Validation procedure
The stimulus validation procedure was based on previous work (*Li et al., 2020*; see *Figure 2—figure supplement 1* for an example of the task). Across nine trials, participants rated the similarity of each of the three shapes in the context of every other shape, as well as four control trials in which each shape was rated relative to itself. For this initial stimulus validation experiment, we used line drawings of the three shapes (for the 4-day crossmodal learning task we used images of the printed objects). Afterwards, participants completed 40 trials in which they rated the similarity of each of the 5 sounds in the context of every other sound, as well as 4 trials in which every sound was rated relative to itself.

In a self-timed manner, participants viewed pictures of shapes or clicked icons to play the to-be-rated sounds from a headset.

For the shapes, we replicated the triangular geometry from participant similarity ratings obtained in our past work (*Li et al., 2020*) indicating that each shape was about as similar as every other shape (*Figure 1a*). We then selected the three sounds that were best equated in terms of their perceived similarity (*Figure 1a*). Thus, like the shapes, this procedure ensured that subjective similarity for the sounds was explicitly controlled but the underlying auditory dimensions could vary (e.g. timbre, pitch, frequency). This initial validation experiment ensured that the subjective similarity of the three features of each stimulus modality was equated within each modality prior to the primary 4-day learning task.

### 3D-printed shape-sound objects

The three validated shapes were 3D-printed using a DREMEL Digilab 3D Printer 3D45-01 with 1.75 mm gold-colored polymerized lactic acid filament. To create the 3D object models, the original 2D images were imported into Blender and elongated to add depth. The face of the shape image created a detachable lid, with a small circular opening to allow wiring to extend to a playable button positioned on the exterior of the shape. An empty space was formed inside the 3D shape for the battery-powered embedded speaker. To ensure that the objects were graspable, each shape was 3D-printed to be approximately the size of an adult hand (*Figure 1c*). The lid of the shape was detached before each learning day (*Figure 2*), with the embedded speaker programmed to play either no sound (Day 1) or to play the paired sound that formed the congruent object (Day 3; *Figure 1a*). After the speaker was programmed, the lid of the shape was reattached using thermoplastic adhesive.

The sounds were played at an audible volume by the 3D-printed shapes during the learning task (see next section). During the scanning sessions, we individually tailored the volume until the participant could hear the sounds clearly when inside the MRI scanner.

### Four-day crossmodal object learning task

#### Participants

Twenty new participants (Females = 13, $M_{age}$ = 23.15 years) were recruited and scanned at the Toronto Neuroimaging Facility. All participants were right-handed, with normal or corrected-to-normal vision, normal hearing, and no history of psychiatric illness. Of the 20 scanned participants, 1 participant dropped out after the first neuroimaging session. Severe distortion was observed in a second participant from a metal retainer and data from this participant was excluded from subsequent analyses. Due to technical difficulties, the functional localizer scans were not saved for one participant and most feature runs could not be completed for a second participant. Overall, the within-subject analyses described in the main text included data from a minimum of 16 participants, with most analyses containing data from 17 participants. Critically, this within-subject learning design increases power to detect an effect.

Compensation was $250 CAD for the two neuroimaging sessions and two behavioral sessions (~ 6 hr total, which included set-up, consent, and debriefing), with a $50 CAD completion bonus.

#### Behavioral tasks

On each behavioral day (Day 1 and Day 3; *Figure 2*), participants completed the following tasks, in this order: Exploration Phase, one Unimodal Feature 1-back run (26 trials), Exploration Phase, one Crossmodal 1-back run (26 trials), Exploration Phase, Pairwise Similarity Task (24 trials), Exploration Phase, Pairwise Similarity Task (24 trials), Exploration Phase, Pairwise Similarity Task (24 trials), and finally, Exploration Phase. To verify learning on Day 3, participants also additionally completed a Learning Verification Task at the end of the session. Details on each task are provided below.

The overall procedure ensured that participants extensively explored the unimodal features on Day 1 and the crossmodal objects on Day 3. The Unimodal Feature and the Crossmodal Object 1-back runs administered on Day 1 and Day 3 served as practice for the neuroimaging sessions on Day 2 and Day 4, during which these 1-back tasks were completed. Each behavioral session required less than 1 hr of total time to complete.

## Day 1 exploration phase

On Day 1 (*Figure 2a*), participants separately learned the shape and sound features in a random order. The 3D shapes were explored and physically palpated by the participants. We also encouraged participants to press the button on each shape, although the button was not operational on this day. Each 3D-printed shape was physically explored for 1 min and each sound was heard through a headset seven times. There were six exploration phases in total, interleaved between the 1-back and pairwise similarity tasks (order provided above). This procedure ensured that each individual stimulus was experienced extensively by the end of the first day.

## Day 3 exploration phase

On Day 3 (*Figure 2c*), participants experienced the 3D-printed shape-sound objects in a random order. The sound was played over the embedded speakers by pressing the now-operational button on each object. Participants were allotted 1 min to physically explore and palpate each shape-sound object, as well as to listen to the associated sound by pressing the button. Like Day 1, there were six exploration phases in total, interleaved between the 1-back and pairwise similarity tasks.

## Pairwise similarity task

Using the same task as the stimulus validation procedure (*Figure 2—figure supplement 1*), participants provided similarity ratings for all combinations of the three validated shapes and three validated sounds (each of the six features were rated in the context of every other feature in the set, with four repeats of the same feature, for a total of 72 trials). More specifically, three stimuli were displayed on each trial, with one at the top and two at the bottom of the screen in the same procedure as we have used previously (*Li et al., 2020*). The 3D shapes were visually displayed as a photo, whereas sounds were displayed on screen in a box that could be played over headphones when clicked with the mouse. The participant made an initial judgment by selecting the more similar stimulus on the bottom relative to the stimulus on the top. Afterwards, the participant made a similarity rating between each bottom stimulus with the top stimulus from 0 being no similarity to five being identical. This procedure ensured that ratings were made relative to all other stimuli in the set.

## Unimodal feature and crossmodal object 1-back tasks

During fMRI scanning on Days 2 and 4, participants completed 1-back tasks in which the target was an exact sequential repeat of a feature (Unimodal Feature Task) or an exact sequential repeat of the shape-sound object (Crossmodal Object Task). In total, there were 10 Unimodal Feature runs and 5 Crossmodal Object runs for each scanning session. Two Unimodal Feature runs were followed by one Crossmodal Object run in an interleaved manner to participants until all 10 Unimodal Feature runs and 5 Crossmodal Object runs were completed. Each run lasted 3 min and had 26 trials.

Each Unimodal Feature and Crossmodal Object run began with a blank screen appearing for 6 s. For Unimodal Feature runs, either a shape or sound feature would then be presented for 2 s, followed by a fixation cross appearing for 2–8 s (sampled from the following probability distribution: 2 s=30%, 4 s=30%, 6 s=30%, and 8 s=10%). For Crossmodal Object runs, each shape appeared on the monitor at the same time as a sound was played through the headset for two seconds, followed by a fixation cross appearing for 2–8 s (sampled from the following probability distribution: 2 s=30%, 4 s=30%, 6 s=30%, and 8 s=10%). Ensuring equal trial numbers, three shape-sound pairings were congruent (learned by participants) and three shape-sound pairings were incongruent (not learned by participants). Congruent and incongruent pairings were built from different combinations of the same shape and sound features, with pairings counterbalanced across participants.

Overall, each stimulus was presented four times in a random order per run, with two repeats occurring at a random position for the corresponding 1-back task. The stimulus identity and temporal position of any given 1-back repeat was random.

## Learning verification task (Day 3 Only)

As the final task on Day 3, participants completed a task to ensure that participants successfully formed their crossmodal pairing. All three shapes and sounds were randomly displayed in six boxes on a display. Photos of the 3D shapes were shown, and sounds were played by clicking the box with

the mouse cursor. The participant was cued with either a shape or sound, and then selected the corresponding paired feature. At the end of Day 3, we found that all participants reached 100% accuracy on this task (10 trials).

## Behavioral pattern similarity analysis

The pairwise similarity ratings for each stimulus were averaged into a single feature-level RDM. We examined the magnitude of pattern similarity for congruent features compared to incongruent features across learning days (see *Figure 2—figure supplement 1*).

### Neuroimaging procedures

Scanning was conducted using a 32-channel receiver head coil with the Siemens Magnetom Prisma 3T MRI scanner at the Toronto Neuroimaging Facility. To record responses, participants used a 4-button keypad (Current Designs, HHSC-1X4 CR). Stimulus materials were displayed using an MR compatible screen at high resolution (1920x1080) with zero-delay timing (32" BOLD screen) controlled by Psych-Toolbox-3 in MATLAB. At the start of each neuroimaging session, we performed a sound check with a set of modified in-ear MR-compatible headphones (Sensimetrics, model S14), followed by a functional localizer and then by the task-related runs.

While in the scanner, participants completed the following: After an initial functional localizer, we collected a resting state scan. After five 1-back runs, we acquired a whole-brain high-resolution T1-weighted structural image. After an additional five 1-back runs, we acquired a second resting-state scan, followed by the last five 1-back runs. The 15 total 1-back runs were interleaved such that 2 Unimodal Feature runs would be presented, followed by 1 Crossmodal Feature run until all 15 runs had been completed (see *Figure 2*).

## Multi-echo fMRI

A 3D multi-echo echo-planer imaging (EPI) sequence with blipped-controlled aliasing in parallel imaging (CAIPI) sampling (*Stirnberg and Stöcker, 2021*) was used to acquire fMRI data on Day 2 and Day 4. For task-related scans, the 3 echoes (TR = 2000ms, TE 1=11ms, TE 2=31.6ms, and TE 3=52.2ms) were each acquired with 90 images (210x210 field of view with a 100x100 matrix resize; anterior to posterior phase encoding, 78 slices, slice thickness: 2.10 mm, flip angle: 17°, interleaved multi-slice acquisition), resulting in an in-plane resolution of 2.10x2.10 mm. 3D distortion correction and pre-scan normalization was enabled, with acceleration factor PE = 2 and acceleration factor 3D=3. These parameters yielded coverage over the entire cortex, and a B0 field map was collected at the completion of the experiment.

### 1-back tasks (unimodal feature runs and crossmodal object runs)

Rather than collecting data from many different instances of a category as is common in a fMRI study using multivariate pattern analysis, we collected data from many repetitions of the *same* stimulus using a psychophysics-inspired approach. This paradigm ensured that the neural representations specific to each unimodal feature and each crossmodal object was well-powered for subsequent pattern similarity analyses (*Coutanche and Thompson-Schill, 2012*). Excluding 1-back repeats, each unimodal feature was displayed four times per run for a total of 40 instances per scanning session (80 instances of each unimodal feature in total). Excluding 1-back repeats, each shape-sound pairing was displayed four times per run for a total of 20 instances per scanning session (40 instances of each shape-sound object in total). We designed our task-related runs to be 3 min in length, as 'mini-runs' have been shown to improve data quality in multivariate pattern analysis (*Coutanche and Thompson-Schill, 2012*). Details of the task can be found in the section above.

### Standard functional localizer

Participants viewed intact visual features and phase scrambled versions of the same features in separate 24 s blocks (8 functional volumes; *Malach et al., 1995*). Each of the 32 images within a block were presented for 400ms each with a 350ms ISI. There were two groups of four blocks, with each group separated by a 12 s fixation cross. Block order was counterbalanced across participants. All stimuli were presented in the context of an 1-back task, and the order of images within blocks was

randomized with the 1-back repeat occurring once per block. The identity and temporal position of the 1-back repeat was random.

### Structural and resting state scans

A standard whole-brain high-resolution T1-weighted structural image was collected (TR = 2000ms, TE = 2.40ms, flip angle = 9°, field of view = 256 mm, 160 slices, slice thickness = 1.00 mm, acceleration factor PE = 2), resulting in an in-place resolution of 1.00 mm x 1.00.

Two 6 min 42 s resting state scans were also collected (TR = 2000ms, TE = 30ms; field of view: 220 mm, slice thickness: 2.00 mm; interleaved multi-slice acquisition, with acceleration factor PE = 2).

## Neuroimaging analysis

### ROI Definitions

We conducted region-of-interest univariate (*Figure 3c and d*) and multivariate pattern analysis (*Figures 4–6*) in five a priori masks: temporal pole, perirhinal cortex, lateral occipital complex (LOC), primary visual cortex (V1), and primary auditory cortex (A1). These regions were selected a priori given their hypothesized role in representing individual unimodal features as well as their integrated whole (*Patterson et al., 2007*; *Cowell et al., 2019*). More specifically, we expected that the anterior temporal lobe structures – temporal pole and perirhinal cortex – would differentiate between the congruent and incongruent conditions. By contrast, we expected LOC, V1, and A1 to possess modality-specific biases for either the visual or sound features. Temporal pole, V1, and A1 masks were extracted from the Harvard-Oxford atlas. The perirhinal cortex mask was created from the average of 55 manually segmented T1 images from a previous publication (*Ritchey et al., 2015*). The LOC mask was extracted from the top 500 voxels in the lateral occipital region of each hemisphere that activated more strongly to intact than phase scrambled objects in the functional localizer (uncorrected voxel-wise p<0.001; *Malach et al., 1995*).

Additionally, we conducted region-of-interest univariate and multivariate pattern analysis in two *exploratory* masks: hippocampus and inferior parietal lobes (*Figure 5—figure supplement 1*). These regions were selected given their hypothesized role in the convergence of unimodal feature representations (*Butler and James, 2011*; *Clouter et al., 2017*; *Viganò and Piazza, 2020*; *Viganò et al., 2021*; *Binder and Desai, 2011*).

Probabilistic masks were thresholded at .5 (i.e. voxels labelled in 50% of participants), with the masks transformed to subject space through the inverse warp matrix generated from FNIRT nonlinear registration (see *Preprocessing*) then resampled from 1mm$^3$ to 2.1 mm$^3$. All subsequent analyses were conducted in subject space.

## Multi-echo ICA-based denoising

For a detailed description of the overall ME-ICA pipeline, see the *tedana* Community (*tedana Community, 2021*). The multi-echo ICA-based denoising approach was implemented using the function *meica.py* in AFNI. We optimally averaged the three echoes, which weights the combination of echoes based on the estimated $T_2^*$ at each voxel for each echo. PCA then reduced the dimensionality of the optimally combined dataset and ICA decomposition was applied to remove non-BOLD noise. TE-dependent components reflecting BOLD-like signal for each run were used as the dataset for subsequent preprocessing in FSL (e.g. see *Figure 3—figure supplement 1*).

## Preprocessing

First, the anatomical image was skull-stripped. Data were high-pass temporally filtered (50 s) and spatially smoothed (6 mm). Functional runs were registered to each participant's high-resolution MPRAGE image using FLIRT boundary-based registration, with registration further refined using FNIRT nonlinear registration. The resulting data were analyzed using first-level FEAT Version 6.00 in each participant's native anatomical space.

## Univariate analysis

To obtain participant-level contrasts, we averaged the run-level Unimodal Feature (*Visual* vs. *Sound*) and Crossmodal Object (*Congruent* vs. *Incongruent*) runs to produce the whole-brain group-level

contrasts in FSL FLAME. Whole-brain analyses were thresholded at voxel-level p=0.001 with random field theory cluster correction at p=0.05.

For ROI-based analyses (*Figure 3*), we estimated percent signal change using *featquery*. The parameter estimates (beta weight) were scaled by the peak height of the regressor, divided by the baseline intensity in the *Visual* vs. *Sound* and *Congruent* vs. *Incongruent* contrasts to obtain a difference score. Inferential statistical analyses were performed with these difference scores using a linear mixed model which included learning day (before vs. after crossmodal learning) and hemisphere (left or right) as fixed effects for each ROI, with participants modelled as random effects. All linear mixed model analyses were conducted using the *nlme* package in R version 3.6.1.

### Single-trial estimates

We used the least squares single approach (*Mumford et al., 2014*) with 2 mm smoothing on the raw data in a separate set of analyses distinct from the univariate contrasts. Each individual stimulus, all other repetitions of the stimulus, and all other individual stimuli were modelled as covariates, allowing us to estimate whole-brain single-trial betas for each trial by run by mask by hemisphere by subject. All pattern similarity analyses described in the main text were conducted using the *CoSMoMVPA* package in MATLAB. After the single-trial betas were estimated, the voxel-wise activity across runs were averaged into a single overall matrix.

### Neuroimaging pattern similarity analysis

Four comparisons were conducted for each a priori ROI: (1) the autocorrelation of the average voxel-wise matrix during Unimodal Feature runs (*Figure 4a*, *Figure 5—figure supplement 1*; *Figure 5—figure supplement 2* ) the correlation between the RDM created from the Unimodal Feature runs before crossmodal learning to the RDM created from the Crossmodal Object runs before crossmodal learning (*Figure 5a*), (3) the correlation between the RDM created from the Unimodal Feature runs before crossmodal learning to the RDM created from the Crossmodal Object runs after crossmodal learning (*Figure 5b*), and (4) the correlation between the RDM created from the Crossmodal Object runs before crossmodal learning to the RDM created from the Crossmodal Object runs after crossmodal learning (*Figure 6*).

The z-transformed Pearson's correlation coefficient was used as the distance metric for all pattern similarity analyses. More specifically, each individual Pearson correlation was Fisher z-transformed and then averaged (see *Corey et al., 1998*). Inferential statistical analyses were performed for each individual ROI using linear mixed models which could include congruency (congruent or incongruent), learning day (before or after crossmodal learning), modality (visual or sound), and hemisphere (left or right) as fixed factors, with participant modelled as random effects allowing intercepts to vary by learning day when appropriate. One-sample t-tests also compared the z-transformed pattern similarity scores relative to 0. All linear mixed model analyses were conducted using the *nlme* package in R version 3.6.1.

### Crossmodal object learning task: behavioral replication

#### Participants

Forty-four new participants (Females = 34, $M_{age}$ = 23.95 years) were recruited from the University of Toronto undergraduate participant pool and from the community. Course credit or $10 /hr CAD was provided as compensation.

#### Procedure

We conducted a same-day behavioral-only variant of the 4-day task described in the main text (*Figure 2*), excluding neuroimaging sessions. Participants first explored the 3D-printed shapes and heard the sounds separately (the button-activated speaker was not operational on this day). Each 3D-printed shape was physically explored for 1 min and each sound was heard through a headset seven times. On a separate pairwise similarity rating task, participants then provided similarity ratings for all combinations of the three shapes and three sounds (rated in the context of each other stimulus in the set, with four repeats of the same item; 72 total trials). Every 24 trials, participants again explored the same shapes and sounds (separately before crossmodal learning, in a counterbalanced order across participants).

Next, participants learned that certain shapes are associated with certain sounds, such that the 3D-printed shapes now played a sound when the button was pressed. Participants were allotted 1 min to physically explore and palpate each shape-sound object, as well as to listen to the associated sound by pressing the button. Participants repeated the pairwise similarity rating task, and every 24 trials, participants explored the 3D-printed shape-sound objects.

The behavioral similarity judgments before and after crossmodal learning were analyzed in the same pattern similarity approach described in the main text (*Figure 2—figure supplement 1*).

## Acknowledgements

We are grateful to the Toronto Neuroimaging community for helpful feedback. In particular, the first author thanks Dr. Katherine Duncan and Dr. Massieh Moayedi for suggestions related to the experimental design, Dr. Michael Mack for initial guidance with 3D-printing, Dr. Rosanna Olsen for her tutorial on medial temporal lobe segmentation, as well as Dr. Andy Lee and Dr. Adrian Nestor for their neuroimaging and multivariate pattern analysis courses. We thank Annie Kim and Katarina Savel for their assistance with participant recruitment, as well as Priya Abraham for her assistance with MRI scanning. Finally, we thank Dr. Rüdiger Stirnberg for sharing with us the multi-echo fMRI sequence used in this manuscript. AYL is supported by an Alexander Graham Bell Canada Graduate Scholarship-Doctoral from the Natural Sciences and Engineering Research Council of Canada (NSERC CGS-D). This work is supported by a Scholar Award from the James S McDonnell Foundation, an Early Researcher Award from the Ontario Government, an NSERC Discovery grant, and a Canada Research Chair to MDB.

## Additional information

### Competing interests

Morgan D Barense: Reviewing editor, eLife. The other authors declare that no competing interests exist.

### Funding

| Funder | Grant reference number | Author |
|---|---|---|
| Natural Sciences and Engineering Research Council of Canada | Alexander Graham Bell Canada Graduate Scholarship-Doctoral | Aedan Yue Li |
| Natural Sciences and Engineering Research Council of Canada | Discovery Grant (RGPIN-2020-05747) | Morgan D Barense |
| James S. McDonnell Foundation | Scholar Award | Morgan D Barense |
| Canada Research Chairs | | Morgan D Barense |
| Ontario Ministry of Research and Innovation | Early Researcher Award | Morgan D Barense |

The funders had no role in study design, data collection and interpretation, or the decision to submit the work for publication.

### Author contributions

Aedan Yue Li, Conceptualization, Data curation, Software, Formal analysis, Supervision, Funding acquisition, Validation, Investigation, Visualization, Methodology, Writing - original draft, Project administration, Writing – review and editing; Natalia Ladyka-Wojcik, Investigation, Project administration, Writing – review and editing; Heba Qazilbash, Data curation, Validation, Investigation, Project administration, Writing – review and editing; Ali Golestani, Resources, Supervision, Methodology, Writing – review and editing; Dirk B Walther, Resources, Supervision, Writing – review and editing; Chris B Martin, Software, Supervision, Writing – review and editing; Morgan D Barense, Conceptualization,

Resources, Supervision, Funding acquisition, Methodology, Project administration, Writing – review and editing

#### Author ORCIDs
Aedan Yue Li ![ORCID] http://orcid.org/0000-0003-0580-4676
Natalia Ladyka-Wojcik ![ORCID] http://orcid.org/0000-0003-1218-0080
Dirk B Walther ![ORCID] https://orcid.org/0000-0001-8585-9858
Chris B Martin ![ORCID] http://orcid.org/0000-0002-7014-4371

#### Ethics
All experiments described in this study were approved by the University of Toronto Ethics Review Board: 37590. Informed consent was obtained for all participants in the study.

#### Decision letter and Author response
Decision letter https://doi.org/10.7554/eLife.83382.sa1
Author response https://doi.org/10.7554/eLife.83382.sa2

## Additional files

#### Supplementary files
• MDAR checklist

#### Data availability
Anonymized data are available on the Open Science Framework: https://osf.io/vq4wj/. Univariate maps are available on NeuroVault: https://neurovault.org/collections/LFDCGMAY/.

The following datasets were generated:

| Author(s) | Year | Dataset title | Dataset URL | Database and Identifier |
|---|---|---|---|---|
| Li AY, Ladyka-Wojcik N, Qazilbash H, Golestani A, Bernhardt-Walther D, Martin CB, Barense MD | 2024 | Experience transforms crossmodal object representations in the anterior temporal lobes | https://doi.org/10.17605/OSF.IO/VQ4WJ | Open Science Framework, 10.17605/OSF.IO/VQ4WJ |
| Li AY, Ladyka-Wojcik N, Qazilbash H, Golestani A, Bernhardt-Walther D, Martin CB, Barense MD | 2024 | Multimodal object representations rely on integrative coding | https://identifiers.org/neurovault.collection:12807 | NeuroVault, 12807 |

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
