## [Editor Report]

The fMRI study is important because it investigates fundamental questions about the neural basis of multimodal binding using an innovative multi-day learning approach. The results provide solid evidence for learning-related changes in the anterior temporal lobe. This paper is of interest to a broad audience of cognitive neuroscientists.

---

## [Decision Letter]

**Decision letter after peer review:**

Thank you for submitting your article "Multimodal Object Representations Rely on Integrative Coding" for consideration by *eLife*. Your article has been reviewed by 3 peer reviewers, one of whom is a member of our Board of Reviewing Editors, and the evaluation has been overseen by Chris Baker as the Senior Editor. The following individuals involved in the review of your submission have agreed to reveal their identity: Sarah Solomon (Reviewer #2), Tim Rogers (Reviewer #3).

As you will read below, the reviewers were enthusiastic about the research question and the design of the study. However, they were not convinced that the current set of results provide strong evidence for the main conclusions of an "explicit integrative representation". They have suggested alternative interpretations and additional analyses that I would encourage you to consider.

Essential revisions (for the authors):

1) Clarify what is meant by an "explicit integrative" representation and improve the motivation for the individual analyses as tests of such a representation. For example, consider including, for each analysis, specific predictions that follow from the main hypothesis (and not alternative hypotheses) of an explicit integrative representation. A schematic illustration may also be helpful.

2) Discuss alternative explanations for the results; the reviewers were not convinced that all analyses provide evidence for an explicit integrative representation.

3) Run additional analyses that may provide more direct evidence for an integrative code.

4) Correct for multiple comparisons, where appropriate, and test the specificity of results to ROIs (e.g., in an ANOVA that includes ROI).

5) Consider including additional ROIs (e.g., ventral ATL/anterior fusiform, hippocampus, parietal cortex).

6) Include additional detail about the procedures and analyses.

*Reviewer #1 (Recommendations for the authors):*

The Introduction could discuss additional literature that is relevant to the current study. For example, previous studies investigated learning associations between modalities, and discussed several ways in which this could lead to neural changes; for example, unisensory stimulation could activate representations of the other modality after learning (Shams and Seitz, TiCS 2008). Previous fMRI work on conceptual/multimodal representations in ATL is not reviewed in much detail. Studies that seem particularly relevant are Vigano and Piazza (2020) and Vigano et al. (2021).

– The focus is specifically on the anterior temporal lobes. However, there are other regions that have been proposed (and shown) to similarly represent multimodal object representations, including the parietal cortex (Binder et al., TiCS 2011). Indeed, Binder et al. write: "the most anterior parts of the temporal lobe, including the TP and anteromedial temporal regions, are unlikely to be a critical hub for retrieval of multimodal semantic knowledge." Discussing these alternative views would be relevant for a balanced article.

– Throughout the manuscript, the comparison is made with knowledge of a frog. While this is useful in the Introduction as general background, to set up the study, I found it somewhat confusing in the Results section. In the current study, the sounds and shapes are arbitrary and are learned for a short period. The resulting neural representation may well be different from the semantic representations of animal shapes and sounds, which have developed over many years.

– The results are interpreted as evidence for an integrative code already in the Results section. For example, on p.9, the incongruent>congruent activation in TP is interpreted as "the formation of an explicit integrative code in the temporal pole". However, the evidence for such a multimodal object code is very indirect here, and there are many alternative explanations for this (and other) findings (e.g., the novelty signal mentioned on p.11). I think it would be better to stay closer to the data when first presenting these individual results. Alternatively, to remain open to alternative interpretations, results can be described as being "consistent with" the integrative code interpretation.

– Relatedly, it was often not clear to me why specific results provide evidence for an integrative code. It might help if you could motivate the analyses in the context of searching for an integrative code, i.e., make explicit predictions based on the hypothesis of an integrative code and then present the results. In this way, the reader can follow the logic of the predictions before being presented with possibly complex results.

– The strongest effect of multimodal learning on univariate activity (Figure 3d) is the reduced visual bias in LOC. This result is not investigated further and does not feature in the interpretation and conclusions. However, this effect seems important to understand, also in relation to a similar (smaller) change in PRC, which is interpreted extensively. Do both these effects reflect the same underlying change in processing the stimuli after learning? Could it reflect visual imagery after multimodal learning? E.g., participants may imagine the sound-associated object after having learned the association, thus making the response evoked by the sound more visual.

– The analysis on page 12 (Figure 4) shows the reduced similarity of associated shapes and sounds in TP after learning. This is an interesting finding but I find it hard to interpret, also in light of the behavioral results showing that the shapes and sounds are perceived as more similar after learning – where is this similarity neurally represented? Could this reflect suppression of the associated sound/shape? Furthermore, I did not understand how this finding provides evidence for an explicit integrative code, with "the whole being different from the parts", considering that this analysis only involved responses to individual modalities.

– The reporting of the analysis on page 13 (Figure 5) differs from how the other analyses are reported, starting by showing an interaction with ROI to motivate only testing one ROI. However, the interaction is between modality and ROI after learning but I suppose this interaction is equally strong before learning. This analysis does not test whether the visual bias in PRC (or other regions) was impacted by learning, yet this is what is concluded: "the PRC was the only region that differed in its modality-specific bias across learning days". A subsequent analysis then tests this more directly, but only for PRC.

– I didn't understand why having a visual or sound bias (in Figure 5) is strong evidence for "an explicit integrative object representation transformed from the original features" (p.14). The specific change in PRC is hard to interpret, considering that the similarity is generally very low and after learning the PRC no longer resembled either of the components. This could reflect increased noise, for example, because patterns are now compared across sessions (see also next point).

– The analysis in Figure 6 shows pattern similarity across days. PRC shows a significant difference before vs after learning but the overall similarity is very low (and not above 0), and the effect is driven by below-zero similarity in the incongruent condition. I'm not sure how a below-zero similarity can be interpreted. The main effect of the region is not very informative here; it does not show that the learning-related difference was specific to PRC.

– A more direct test of "the whole is different from the sum of the parts" would be to model the combined response using the individual feature responses (e.g., see Baldassano et al., Cereb Cortex 2017). You would expect that a linear combination of sound and shape activity patterns (e.g., the average) is less similar to congruent objects than incongruent objects after learning. Including this test would be important to address the key hypothesis.

*Reviewer #2 (Recommendations for the authors):*

– The only explanation of the theory of "explicit integrative representations" is that the "whole is different than the sum of its parts." This is not enough for the reader to fully understand the manuscript's theoretical claims. An explanation of what is meant by "explicit" and what is meant by "integrative" seems necessary since researchers in different fields and subfields might interpret those terms in meaningfully different ways. If the claim is that "explicit integrative representations" are abstracted away from featural information, the currently presented analyses are not convincing. If the claim is that object representations contain featural information that has been warped or transformed, this is more strongly supported by the presented data. I think the authors are arguing for the former rather than the latter, but the data seem to support the latter over the former. More clarity on the theoretical claims and how the data speak to those claims would be helpful.

– An explanation for the direction of the univariate effect in the temporal pole and acknowledgement of alternative interpretations is warranted. The authors mention in a later section that this result could be a novelty response, but acknowledging this possibility in the same section the data are reported feels pertinent, especially since other interpretations are provided ("these neural changes imply the formation of an explicit integrative code in the temporal pole").

– The finding that congruent visual-sound pairs were more dissimilar after multimodal object learning should be contextualized within the theory of "explicit integrative" representations. First, why would this representational theory predict this direction of representational change? Second, if one's claim is that a new, explicit representation is formed to represent the learned multimodal object, a learning-evoked change in unimodal feature representations seems to contradict that theory. If the explicit object representation is distinct from the features themselves, wouldn't it follow that the unimodal features should remain unchanged? Explaining how the results help discriminate between the two representational theories raised in the introduction, and how it specifically supports the explicit integrative theory, would enable the reader to contextualize the reported findings.

– The finding that unimodal features correlate with their respective multimodal object representations before but not after learning in the perirhinal cortex provides the best support for the claim that object representations are independent of feature representations. However, there are two modifications to the current analysis that could make this argument more direct. First, readers would want to know that unimodal features no longer correlate with congruent objects in the perirhinal cortex, but that their correlations with incongruent objects are unaffected. Otherwise, we can't interpret this result as due to multimodal object learning per se. Second, there's the question of representational stability in the perirhinal cortex. If perirhinal feature representations are not stable across days, it is possible that featural content is actually present in the object representations, but this would only be evident if one used the Day 4 feature representations instead of the Day 2 feature representations. If the Day 4 feature representations do not correlate with the Day 4 congruent object representations, this would be the most direct evidence for explicit, integrative object representations that are distinct from feature representations in the perirhinal cortex.

– There is a conflation between "visual features" and "objects" throughout the manuscript which can be quite confusing. Sometimes the word "object" is used to represent multimodal visual-auditory objects ("multimodal object"), other times "object" refers to complex visual objects with multiple features ("frog"), and sometimes simple visual stimuli (in functional localizer). In particular, the frog example used throughout the manuscript doesn't feel appropriate, because the representation of "frog" is a lot more complex than one visual feature, and is already an integrated representation across different visual features and modalities (even if the "croak" feature is hypothetically removed). The frog example also muddies the theoretical claims-the authors want the "frog" representation to change when paired with "croak" because "frog" is already an integrated object representation that now needs to be modified. However, the authors should *not* want the representation of a single visual feature to change, since one visual feature is an ingredient that is fed into a separate, integrated object representation.

*Reviewer #3 (Recommendations for the authors):*

I would encourage the authors to provide open access to the data and analysis code.

---

## [Author Response]

Essential revisions (for the authors):1) Clarify what is meant by an "explicit integrative" representation and improve the motivation for the individual analyses as tests of such a representation. For example, consider including, for each analysis, specific predictions that follow from the main hypothesis (and not alternative hypotheses) of an explicit integrative representation. A schematic illustration may also be helpful.2) Discuss alternative explanations for the results; the reviewers were not convinced that all analyses provide evidence for an explicit integrative representation.3) Run additional analyses that may provide more direct evidence for an integrative code.4) Correct for multiple comparisons, where appropriate, and test the specificity of results to ROIs (e.g., in an ANOVA that includes ROI).5) Consider including additional ROIs (e.g., ventral ATL/anterior fusiform, hippocampus, parietal cortex).6) Include additional detail about the procedures and analyses.

We thank the editor and reviewers for their careful reading of the manuscript and are grateful for the constructive feedback.

Overall, we have rewritten the manuscript based on the reviewer feedback. More specifically, we have now (1) elaborated on why our methodological design allows us to make claims about “crossmodal integrative coding” in anterior temporal lobe structures that is different from the representations of the unimodal features, as well as softened our claims substantially (e.g., new title, rewrote results and discussion), (2) discussed alternative interpretations of the data, (3) performed several additional analyses suggested by the reviewers, (4) clarified why we have not controlled for multiple comparisons, by removing a series of post hoc across-ROI comparisons that were irrelevant to the key questions of the present manuscript, (5), included additional ROIs as suggested by the reviewers, and (6) discussed key analytic choices and include substantially more detail about the procedures and analyses.

Reviewer #1 (Recommendations for the authors):The Introduction could discuss additional literature that is relevant to the current study. For example, previous studies investigated learning associations between modalities, and discussed several ways in which this could lead to neural changes; for example, unisensory stimulation could activate representations of the other modality after learning (Shams and Seitz, TiCS 2008). Previous fMRI work on conceptual/multimodal representations in ATL is not reviewed in much detail. Studies that seem particularly relevant are Vigano and Piazza (2020) and Vigano et al. (2021).– The focus is specifically on the anterior temporal lobes. However, there are other regions that have been proposed (and shown) to similarly represent multimodal object representations, including the parietal cortex (Binder et al., TiCS 2011). Indeed, Binder et al. write: "the most anterior parts of the temporal lobe, including the TP and anteromedial temporal regions, are unlikely to be a critical hub for retrieval of multimodal semantic knowledge." Discussing these alternative views would be relevant for a balanced article.

We thank the reviewer for pointing us to this literature and now cite these papers in the manuscript where appropriate. Vigano and Piazza (2020) examined navigation along dimensions on a conceptual space, finding evidence of direction-based coding in the prefrontal cortex and entorhinal cortex. Interestingly, it is likely that our perirhinal cortex masks overlap with what Vigano and Piazza (2020) had defined as entorhinal cortex. Vigano et. al (2021) found that the parietal cortex may represent absolute direction along conceptual spaces. This is consistent with Binder (2011), who argued that the inferior parietal lobes are essential for the retrieval of multimodal semantic knowledge. Finally, Shams and Seitz described the importance of multisensory learning. Notably, and in contrast to this previous work, our study investigated whether crossmodal representations are entirely constructed from their unimodal features or whether crossmodal representations are distinct from the sum of their component features. While we agree that previous findings have linked multiple brain regions to multisensory representations, this previous work did not consider crossmodal object representations separately from their component unimodal feature-level representations, which was the central focus of our present study. Depending on the research question, factors such as complexity, familiarity with the unimodal features, and subjective similarity may also be uncontrolled, which may complicate the interpretation of results.

Given this, we preferred to keep our introduction focused on our specific question of the relationship between crossmodal object representations and the component unimodal features. However, we agree with both Reviewers 1 and 3 that these other brain regions are interesting as they relate to binding and now include them as exploratory analyses in Supplemental Figure 4. We did not find evidence of crossmodal integrative coding in the inferior parietal lobes or hippocampus. Instead, these regions better seemed to represent the *convergence* of unimodal feature representations.

We now discuss this at various points throughout the manuscript:

“Notably, our perirhinal cortex mask overlaps with a key region of the ventral anterior temporal lobe thought to be the central locus of crossmodal integration in the “hub and spokes” model of semantic representations.^9,50^ However, additional work has also linked other brain regions to the convergence of unimodal representations, such as the hippocampus^51,52,53^ and inferior parietal lobes.^54,55^ This past work on the hippocampus and inferior parietal lobe does not necessarily address the crossmodal binding problem that was the main focus of our present study, as previous findings often do not differentiate between crossmodal integrative coding and the convergence of unimodal feature representations per se. Furthermore, previous studies in the literature typically do not control for stimulus-based factors such as experience with unimodal features, subjective similarity, or feature identity that may complicate the interpretation of results when determining regions important for crossmodal integration. Indeed, we found evidence consistent with the convergence of unimodal feature-based representations in both the hippocampus and inferior parietal lobes (Figure 5 —figure supplement 1), but no evidence of crossmodal integrative coding different from the unimodal features. The hippocampus and inferior parietal lobes were both sensitive to visual and sound features before and after crossmodal learning (see Figure 5 —figure supplement 1). Yet the hippocampus and inferior parietal lobes did not differentiate between the congruent and incongruent conditions or change with experience (see Figure 5 —figure supplement 1).” – pg. 15

“Analyses for the hippocampus (HPC) and inferior parietal lobe (IPL). (a) In the visual vs. auditory univariate analysis, there was no visual or sound bias in HPC, but there was a bias towards sounds that increased numerically after crossmodal learning in the IPL. (b) Pattern similarity analyses between unimodal features associated with congruent objects and incongruent objects. Similar to Figure 4 —figure supplement 1, there was no main effect of congruency in either region. (c) When we looked at the pattern similarity between Unimodal Feature runs on Day 2 to Crossmodal Object runs on Day 2, we found that there was significant pattern similarity when there was a match between the unimodal feature and the crossmodal object (e.g., pattern similarity > 0). This pattern of results held when (d) correlating the Unimodal Feature runs on Day 2 to Crossmodal Object runs on Day 4, and (e) correlating the Unimodal Feature runs on Day 4 to Crossmodal Object runs on Day 4. Finally, (f) there was no significant pattern similarity between Crossmodal Object runs before learning correlated to Crossmodal Object after learning in HPC, but there was significant pattern similarity in IPL (p < 0.001). Taken together, these results suggest that both HPC and IPL are sensitive to visual and sound content, as the (c, d, e) unimodal feature-level representations were correlated to the crossmodal object representations irrespective of learning day. However, there was no difference between congruent and incongruent pairings in any analysis, suggesting that HPC and IPL did not represent crossmodal objects differently from the component unimodal features. For these reasons, HPC and IPL may represent the convergence of unimodal feature representations (i.e., because HPC and IPL were sensitive to both visual and sound features), but our results do not seem to support these regions in forming crossmodal integrative coding distinct from the unimodal features (i.e., because HPC and IPL did not differentiate the congruent and incongruent conditions and did not change with experience). * p < 0.05, ** p < 0.01, *** p < 0.001. Asterisks above or below bars indicate a significant difference from zero. Horizontal lines within brain regions in (a) reflect an interaction between modality and learning day, whereas horizontal lines within brain regions in reflect main effects of (b) learning day, (c-e) modality, or (f) congruency.” – Figure 5 —figure supplement 1.

– Throughout the manuscript, the comparison is made with knowledge of a frog. While this is useful in the Introduction as general background, to set up the study, I found it somewhat confusing in the Results section. In the current study, the sounds and shapes are arbitrary and are learned for a short period. The resulting neural representation may well be different from the semantic representations of animal shapes and sounds, which have developed over many years.

We thank the reviewer for this suggestion. We have now removed the frog examples when describing the results.

– The results are interpreted as evidence for an integrative code already in the Results section. For example, on p.9, the incongruent>congruent activation in TP is interpreted as "the formation of an explicit integrative code in the temporal pole". However, the evidence for such a multimodal object code is very indirect here, and there are many alternative explanations for this (and other) findings (e.g., the novelty signal mentioned on p.11). I think it would be better to stay closer to the data when first presenting these individual results. Alternatively, to remain open to alternative interpretations, results can be described as being "consistent with" the integrative code interpretation.– Relatedly, it was often not clear to me why specific results provide evidence for an integrative code. It might help if you could motivate the analyses in the context of searching for an integrative code, i.e., make explicit predictions based on the hypothesis of an integrative code and then present the results. In this way, the reader can follow the logic of the predictions before being presented with possibly complex results.

We agree with the reviewer that some of our evidence is indirect and have now tried to stay much closer to the data when presenting individual results. In addition, the Results section has now been completely rewritten to provide better motivation and a clearer description of our analyses.

Furthermore, we now clarify that our experimental task equated the unimodal feature representations in the congruent and incongruent conditions. Congruent and incongruent objects have the same features – the only difference between the conditions was the learned crossmodal representation that emerged with experience. Thus, neural differences between the congruent and incongruent conditions would suggest that the crossmodal object representation is sensitive to the combinations of those features, over and above the identity of those features themselves. Finally, we removed the original novelty signal mentioned on pg. 11, because this was referring to the univariate results and may be a source of confusion.

Overall, when presenting individual results we offer minimal interpretation, as well as better describe our key methodological design in the introduction. For example:

“Critically, our four-day learning task allowed us to isolate any neural activity associated with integrative coding in anterior temporal lobe structures that emerges with experience and differs from the neural patterns recorded at baseline. The learned and non-learned crossmodal objects were constructed from the same set of three validated shape and sound features, ensuring that factors such as familiarity with the unimodal features, subjective similarity, and feature identity were tightly controlled (Figure 2). If the mind represented crossmodal objects entirely as the reactivation of unimodal shapes and sounds (i.e., objects are constructed from their parts), then there should be no difference between the learned and non-learned objects (because they were created from the same three shapes and sounds). By contrast, if the mind represented crossmodal objects as something over and above their component features (i.e., representations for crossmodal objects rely on integrative coding that is different from the sum of their parts), then there should be behavioral and neural differences between learned and non-learned crossmodal objects (because the only difference across the objects is the learned relationship between the parts). Furthermore, this design allowed us to determine the relationship between the object representation acquired after crossmodal learning and the unimodal feature representations acquired before crossmodal learning. That is, we could examine whether learning led to abstraction of the object representations such that it no longer resembled the unimodal feature representations.” – pg. 5

“As a central goal of our study was to identify brain regions that were influenced by the learned crossmodal associations, we next examined univariate differences between Congruent vs. Incongruent for crossmodal object runs as a function of whether the crossmodal association had been learned. We conducted a linear mixed model for each ROI which included learning day (before vs. after crossmodal learning) and congruency (congruent vs. incongruent objects) as fixed factors. We observed a significant interaction between learning day and congruency in the temporal pole (F_1,48_ = 7.63, p = 0.0081, η^2^ = 0.14). Critically, there was no difference in activity between congruent and incongruent objects at baseline before crossmodal learning (t_33_ = 0.37, p = 0.72), but there was more activation to incongruent compared to congruent objects after crossmodal learning (t_33_ = 2.42, p = 0.021, Cohen’s d = 0.42). As the unimodal shape-sound features experienced by participants were the same before and after crossmodal learning (Figure 2), this finding reveals that the univariate signal in the temporal pole was differentiated between congruent and incongruent objects that had been constructed from the same unimodal features.” – pg. 8

“By contrast, we did not observe a univariate difference between the congruent and incongruent conditions in the perirhinal cortex, LOC, V1, or A1 (F_1,45-48_ between 0.088 and 2.34, p between 0.13 and 0.77). Similarly, the exploratory ROIs hippocampus (HPC: F_1,48_ = 0.32, p = 0.58) and inferior parietal lobe (IPL: F_1,48_ = 0.094, p = 0.76) did not distinguish between the congruent and incongruent conditions.” – pg. 8

– The strongest effect of multimodal learning on univariate activity (Figure 3d) is the reduced visual bias in LOC. This result is not investigated further and does not feature in the interpretation and conclusions. However, this effect seems important to understand, also in relation to a similar (smaller) change in PRC, which is interpreted extensively. Do both these effects reflect the same underlying change in processing the stimuli after learning? Could it reflect visual imagery after multimodal learning? E.g., participants may imagine the sound-associated object after having learned the association, thus making the response evoked by the sound more visual.

We thank the reviewer for highlighting this issue. We primarily focused on the anterior temporal lobes due to prior theoretical work on the importance of this region to concept representations. As the reviewer mentions, there could be several factors that might be at play – these findings could reflect visual imagery or also feedback connectivity between the anterior temporal lobes to the LOC. We now make the rationale of our study and analyses clearer in the manuscript, for example:

“Thus, a key theoretical challenge central to resolving the crossmodal binding problem is understanding how anterior temporal lobe structures form object representations. Are crossmodal objects entirely built from features distributed across sensory regions, or is there also integrative coding in the anterior temporal lobes? Furthermore, the existing literature has predominantly studied the neural representation of well-established object concepts from the visual domain alone,^8-25^ even though human experience is fundamentally crossmodal.” – pg. 4

“As a central goal of our study was to identify brain regions that were influenced by the learned crossmodal associations, we next examined univariate differences between Congruent vs. Incongruent for crossmodal object runs as a function of whether the crossmodal association had been learned.” – pg. 8

“Importantly, the initial visual shape bias observed in the perirhinal cortex was attenuated by experience (Figure 5, Figure 5 —figure supplement 2), suggesting that the perirhinal representations had become abstracted and were no longer predominantly grounded in a single modality after crossmodal learning. One possibility may be that the perirhinal cortex is by default visually driven as an extension to the ventral visual stream,^10,11,12^ but can act as a polymodal “hub” region for additional crossmodal input following learning. A complementary possibility may be that our visual features contained tactile information (Figure 1c) that the perirhinal cortex may be sensitive to following the initial exploration phase on our task (Figure 2).^40^ Critically, other brain regions like the LOC also reduced in visual bias (Figure 3c), which may reflect visual imagery or feedback connectivity between the anterior temporal lobes. However, the perirhinal cortex was the only region where the visual bias was entirely attenuated following crossmodal learning (Figure 5b).” – pg. 14

– The analysis on page 12 (Figure 4) shows the reduced similarity of associated shapes and sounds in TP after learning. This is an interesting finding but I find it hard to interpret, also in light of the behavioral results showing that the shapes and sounds are perceived as more similar after learning – where is this similarity neurally represented? Could this reflect suppression of the associated sound/shape? Furthermore, I did not understand how this finding provides evidence for an explicit integrative code, with "the whole being different from the parts", considering that this analysis only involved responses to individual modalities.

We thank the reviewer for identifying this source of confusion. We now better clarify that such a finding could be taken as evidence that learning crossmodal object concepts transforms the original representation of the component unimodal features.

“We next conducted a series of representational similarity analyses across Unimodal Feature and Crossmodal Object runs before and after crossmodal learning. Here, we investigated whether representations for unimodal features were changed after learning the crossmodal associations between those features (i.e., learning the crossmodal pairings that comprised the shape-sound objects). Such a finding could be taken as evidence that learning crossmodal object concepts transforms the original representation of the component unimodal features. More specifically, we compared the correlation between congruent and incongruent shape-sound features within Unimodal Feature runs before and after crossmodal learning (Figure 4a).” – pg. 8

In the discussion, we suggest that the overall pattern of results could indicate that the representation in the anterior temporal lobes reflects crossmodal integrative coding, but remain open to alternative interpretations.

“As one solution to the crossmodal binding problem, we suggest that the temporal pole and perirhinal cortex form unique crossmodal object representations that are different from the distributed features in sensory cortex (Figure 4, 5, 6, Figure 5 —figure supplement 2). However, the nature by which the integrative code is structured and formed in the temporal pole and perirhinal cortex following crossmodal experience – such as through transformations, warping, or other factors – is an open question and an important area for future investigation. Furthermore, these anterior temporal lobe structures may be involved with integrative coding in different ways. For example, the crossmodal object representations measured after learning were found to be related to the component unimodal feature representations measured before learning in the temporal pole but not the perirhinal cortex (Figure 5, 6, Figure 5 —figure supplement 2). Furthermore, pattern similarity for congruent shape-sound pairs were lower than the pattern similarity for incongruent shape-sound pairs after crossmodal learning in the temporal pole but not the perirhinal cortex (Figure 4b, Figure 4 —figure supplement 1). As one interpretation of this pattern of results, the temporal pole may represent new crossmodal objects by combining previously learned knowledge. ^8,9,10,11,13,14,15,33^ Specifically, research into conceptual combination has linked the anterior temporal lobes to compound object concepts such as “hummingbird”.^34,35,36^ For example, participants during our task may have represented the sound-based “humming” concept and visually-based “bird” concept on Day 1, forming the crossmodal “hummingbird” concept on Day 3; Figure 1, 2, which may recruit less activity in temporal pole than an incongruent pairing such as “barking-frog”. For these reasons, the temporal pole may form a crossmodal object code based on pre-existing knowledge, resulting in reduced neural activity (Figure 3d) and pattern similarity towards features associated with learned objects (Figure 4b).” – pg. 12

– The reporting of the analysis on page 13 (Figure 5) differs from how the other analyses are reported, starting by showing an interaction with ROI to motivate only testing one ROI. However, the interaction is between modality and ROI after learning but I suppose this interaction is equally strong before learning. This analysis does not test whether the visual bias in PRC (or other regions) was impacted by learning, yet this is what is concluded: "the PRC was the only region that differed in its modality-specific bias across learning days". A subsequent analysis then tests this more directly, but only for PRC.

The reviewer is correct that the interaction is equally strong before and after learning. However, the perirhinal cortex was the only region to lose its visually biased coding across brain regions in a direct interaction before and after learning. Furthermore, the perirhinal cortex was an a priori selected brain region given previous literature relating the anterior temporal lobes to crossmodal integration. Moreover, we presented the results this way for clarity, and now further reword this section:

“To investigate this effect in perirhinal cortex more specifically, we conducted a linear mixed model to directly compare the change in the visual bias of perirhinal representations from before crossmodal learning to after crossmodal learning (green regions in Figure 5a vs. 5b). Specifically, the linear mixed model included learning day (before vs. after crossmodal learning) and modality (visual feature match to crossmodal object vs. sound feature match to crossmodal object). Results revealed a significant interaction between learning day and modality in the perirhinal cortex (F_1,775_ = 5.56, p = 0.019, η^2^ = 0.071), meaning that the baseline visual shape bias observed in perirhinal cortex (green region of Figure 5a) was significantly attenuated with experience (green region of Figure 5b). After crossmodal learning, a given shape no longer invoked significant pattern similarity between objects that had the same shape but differed in terms of what they sounded like. Taken together, these results suggest that prior to learning the crossmodal objects, the perirhinal cortex had a default bias toward representing the visual shape information and was not representing sound information of the crossmodal objects. After crossmodal learning, however, the visual shape bias in perirhinal cortex was no longer present. That is, with crossmodal learning, the representations within perirhinal cortex started to look less like the visual features that comprised the crossmodal objects, providing evidence that the perirhinal representations were no longer predominantly grounded in the visual modality.” – pg. 10

“One theoretical view from the cognitive sciences suggests that crossmodal objects are built from component unimodal features represented across distributed sensory regions.^8^ Under this view, when a child thinks about “frog”, the visual cortex represents the appearance of the shape of the frog whereas the auditory cortex represents the croaking sound. Alternatively, other theoretical views predict that multisensory objects are not only built from their component unimodal sensory features, but that there is also a crossmodal integrative code that is different from the sum of these parts.^9,10,11,12,13^ These latter views propose that anterior temporal lobe structures can act as a polymodal “hub” that combines separate features into integrated wholes.^9,11,14,15^” – pg. 4

– I didn't understand why having a visual or sound bias (in Figure 5) is strong evidence for "an explicit integrative object representation transformed from the original features" (p.14). The specific change in PRC is hard to interpret, considering that the similarity is generally very low and after learning the PRC no longer resembled either of the components. This could reflect increased noise, for example, because patterns are now compared across sessions (see also next point).

We thank the reviewer for highlighting this issue. We now better clarify that Figure 5 indicates that the originally visually-biased representations in perirhinal cortex was changed with experience. More specifically, we found that prior to learning the crossmodal objects, the perirhinal cortex had a default bias toward representing the visual shape information but there was no evidence that perirhinal cortex was tracking the unimodal sound features on the crossmodal objects. After crossmodal learning, the visual shape bias in perirhinal cortex is no longer present – that is, with crossmodal learning, the perirhinal cortex started to look less like the visual features that comprise the crossmodal objects and were no longer predominantly grounded in a single modality.

“Taken together, these results suggest that prior to learning the crossmodal objects, the perirhinal cortex had a default bias toward representing the visual shape information and was not representing sound information of the crossmodal objects. After crossmodal learning, however, the visual shape bias in perirhinal cortex was no longer present. That is, with crossmodal learning, the representations within perirhinal cortex started to look less like the visual features that comprised the crossmodal objects, providing evidence that the perirhinal representations were no longer predominantly grounded in the visual modality.” – pg. 10

“Importantly, the initial visual shape bias observed in the perirhinal cortex was attenuated by experience (Figure 5, Figure 5 —figure supplement 2), suggesting that the perirhinal representations had become abstracted and were no longer predominantly grounded in a single modality after crossmodal learning. One possibility may be that the perirhinal cortex is by default visually driven as an extension to the ventral visual stream,^10,11,12^ but can act as a polymodal “hub” region for additional crossmodal input following learning.” – pg. 14

Notably, these results are unlikely to be driven by noise or poor alignment of patterns across sessions, as not all brain regions decreased in pattern similarity across days, even in regions with pattern similarity numerically like that of the PRC (e.g., TP and A1 did not change in Figure 5, nor did HPC and IPL in Figure 5 —figure supplement 1).

“Importantly, the change in pattern similarity in the perirhinal cortex across learning days (Figure 5) is unlikely to be driven by noise, poor alignment of patterns across sessions, or generally reduced responses. Other regions with numerically similar pattern similarity to perirhinal cortex did not change across learning days (e.g., visual features x crossmodal objects in A1 in Figure 5; the exploratory ROI hippocampus with numerically similar pattern similarity to perirhinal cortex also did not change in Figure 5 —figure supplement 1).” – pg. 11

– The analysis in Figure 6 shows pattern similarity across days. PRC shows a significant difference before vs after learning but the overall similarity is very low (and not above 0), and the effect is driven by below-zero similarity in the incongruent condition. I'm not sure how a below-zero similarity can be interpreted. The main effect of the region is not very informative here; it does not show that the learning-related difference was specific to PRC.

We thank the reviewer for this suggestion. In the discussion, we suggest that one possibility is that this below zero pattern similarity reflects a non-linear transformation, such that crossmodal representations in the incongruent condition are made to be dissimilar from the unimodal features before learning (e.g., anticorrelation), whereas crossmodal representations in the congruent condition are made to be orthogonal from the unimodal features before learning (e.g., no correlation) – suggestive of pattern separation. Another possibility – as Reviewer 2 suggested – is that incongruent representations are less stable, with the representation warped to a greater extent than the congruent objects. Future work should more directly explore the structure of the integrative code that emerges with experience (as previously discussed in the response to reviewers and on page 13).

“By contrast, perirhinal cortex may be involved in pattern separation following crossmodal experience. In our task, participants had to differentiate congruent and incongruent objects constructed from the same three shape and sound features (Figure 2). An efficient way to solve this task would be to form distinct object-level outputs from the overlapping unimodal feature-level inputs such that congruent objects are made to be orthogonal from the representations before learning (i.e., measured as pattern similarity equal to 0 in the perirhinal cortex; Figure 5b, 6, Figure 5 —figure supplement 2), whereas non-learned incongruent objects could be made to be dissimilar from the representations before learning (i.e., anticorrelation, measured as patten similarity less than 0 in the perirhinal cortex; Figure 6). Because our paradigm could decouple neural responses to the learned object representations (on Day 4) from the original component unimodal features at baseline (on Day 2), these results could be taken as evidence of pattern separation in the human perirhinal cortex.^11,12^ However, our pattern of results could also be explained by other types of crossmodal integrative coding. For example, incongruent object representations may be less stable than congruent object representations, such that incongruent objects representation are warped to a greater extent than congruent objects (Figure 6).” – pg. 13

– A more direct test of "the whole is different from the sum of the parts" would be to model the combined response using the individual feature responses (e.g., see Baldassano et al., Cereb Cortex 2017). You would expect that a linear combination of sound and shape activity patterns (e.g., the average) is less similar to congruent objects than incongruent objects after learning. Including this test would be important to address the key hypothesis.

We thank the reviewer for this suggestion. We conducted this analysis, but there was no significant difference between congruent or incongruent objects when correlated to an average of sound and shape features in any brain region (within days or across days). However, this could be because participants are representing both the congruent and incongruent objects differently from the unimodal features. That is, both the congruent and incongruent objects were transformed with experience and different from the unimodal features in anterior temporal lobe structures. By contrast, a version of this analysis was significant in the temporal pole (shown in Figure 4b), whereby the features associated with congruent objects are less similar than the features associated with incongruent objects.

Moreover, we softened claims throughout the manuscript regarding the “whole is different from the sum of the parts”. However, we suggest that aspects of our results are consistent with this interpretation:

“Within the perirhinal cortex, the acquired crossmodal object concepts (measured after crossmodal learning) became less similar to their original component unimodal features (measured at baseline before crossmodal learning); Figure 5, 6, Figure 5 —figure supplement 2. This is consistent with the idea that object representations in perirhinal cortex integrate the component sensory features into a whole that is different from the sum of the component parts, which might be a mechanism by which object concepts obtain their abstraction.” – pg. 12

Although the current work provides evidence for crossmodal integrative coding in the anterior temporal lobes, the structure of how this integrative coding emerges with experience – through transformations, warping, or another factor – is an open question and an important area for future investigation.

“As one solution to the crossmodal binding problem, we suggest that the temporal pole and perirhinal cortex form unique crossmodal object representations that are different from the distributed features in sensory cortex (Figure 4, 5, 6, Figure 5 —figure supplement 2). However, the nature by which the integrative code is structured and formed in the temporal pole and perirhinal cortex following crossmodal experience – such as through transformations, warping, or other factors – is an open question and an important area for future investigation.” – pg. 12

Reviewer #2 (Recommendations for the authors):– The only explanation of the theory of "explicit integrative representations" is that the "whole is different than the sum of its parts." This is not enough for the reader to fully understand the manuscript's theoretical claims. An explanation of what is meant by "explicit" and what is meant by "integrative" seems necessary since researchers in different fields and subfields might interpret those terms in meaningfully different ways. If the claim is that "explicit integrative representations" are abstracted away from featural information, the currently presented analyses are not convincing. If the claim is that object representations contain featural information that has been warped or transformed, this is more strongly supported by the presented data. I think the authors are arguing for the former rather than the latter, but the data seem to support the latter over the former. More clarity on the theoretical claims and how the data speak to those claims would be helpful.

We thank the reviewer for these suggestions. We have clarified that we mean “cross-modal integrative coding” to be a whole object representation that is different from the original unimodal features (pg. 5). We have also now removed the term “explicit”.

“Critically, our four-day learning task allowed us to isolate any neural activity associated with integrative coding in anterior temporal lobe structures that emerges with experience and differs from the neural patterns recorded at baseline. The learned and non-learned crossmodal objects were constructed from the same set of three validated shape and sound features, ensuring that factors such as familiarity with the unimodal features, subjective similarity, and feature identity were tightly controlled (Figure 2). If the mind represented crossmodal objects entirely as the reactivation of unimodal shapes and sounds (i.e., objects are constructed from their parts), then there should be no difference between the learned and non-learned objects (because they were created from the same three shapes and sounds). By contrast, if the mind represented crossmodal objects as something over and above their component features (i.e., representations for crossmodal objects rely on integrative coding that is different from the sum of their parts), then there should be behavioral and neural differences between learned and non-learned crossmodal objects (because the only difference across the objects is the learned relationship between the parts). Furthermore, this design allowed us to determine the relationship between the object representation acquired after crossmodal learning and the unimodal feature representations acquired before crossmodal learning. That is, we could examine whether learning led to abstraction of the object representations such that it no longer resembled the unimodal feature representations.” – pg. 5

Furthermore, we have removed our interpretations from the Results section, and instead provide our suggested interpretation in the Discussion section. We highlight that while we find crossmodal integrative coding that is different from the unimodal feature representations in anterior temporal lobe structures, the structure of how this integrative code emerges with experience is less clear and is an important avenue of future research (pg. 18).

“By contrast, perirhinal cortex may be involved in pattern separation following crossmodal experience. In our task, participants had to differentiate congruent and incongruent objects constructed from the same three shape and sound features (Figure 2). An efficient way to solve this task would be to form distinct object-level outputs from the overlapping unimodal feature-level inputs such that congruent objects are made to be orthogonal from the representations before learning (i.e., measured as pattern similarity equal to 0 in the perirhinal cortex; Figure 5b, 6, Figure 5 —figure supplement 2), whereas non-learned incongruent objects could be made to be dissimilar from the representations before learning (i.e., anticorrelation, measured as patten similarity less than 0 in the perirhinal cortex; Figure 6). Because our paradigm could decouple neural responses to the learned object representations (on Day 4) from the original component unimodal features at baseline (on Day 2), these results could be taken as evidence of pattern separation in the human perirhinal cortex.^11,12^ However, our pattern of results could also be explained by other types of crossmodal integrative coding. For example, incongruent object representations may be less stable than congruent object representations, such that incongruent objects representation are warped to a greater extent than congruent objects (Figure 6).” – pg. 13

– An explanation for the direction of the univariate effect in the temporal pole and acknowledgement of alternative interpretations is warranted. The authors mention in a later section that this result could be a novelty response, but acknowledging this possibility in the same section the data are reported feels pertinent, especially since other interpretations are provided ("these neural changes imply the formation of an explicit integrative code in the temporal pole").

We thank the reviewer for this suggestion and now temper our wording when describing the univariate results in the temporal pole.

“As a central goal of our study was to identify brain regions that were influenced by the learned crossmodal associations, we next examined univariate differences between Congruent vs. Incongruent for crossmodal object runs as a function of whether the crossmodal association had been learned. We conducted a linear mixed model for each ROI which included learning day (before vs. after crossmodal learning) and congruency (congruent vs. incongruent objects) as fixed factors. We observed a significant interaction between learning day and congruency in the temporal pole (F_1,48_ = 7.63, p = 0.0081, η^2^ = 0.14). Critically, there was no difference in activity between congruent and incongruent objects at baseline before crossmodal learning (t_33_ = 0.37, p = 0.72), but there was more activation to incongruent compared to congruent objects after crossmodal learning (t_33_ = 2.42, p = 0.021, Cohen’s d = 0.42). As the unimodal shape-sound features experienced by participants were the same before and after crossmodal learning (Figure 2), this finding reveals that the univariate signal in the temporal pole was differentiated between congruent and incongruent objects that had been constructed from the same unimodal features.” – pg. 8

– The finding that congruent visual-sound pairs were more dissimilar after multimodal object learning should be contextualized within the theory of "explicit integrative" representations. First, why would this representational theory predict this direction of representational change? Second, if one's claim is that a new, explicit representation is formed to represent the learned multimodal object, a learning-evoked change in unimodal feature representations seems to contradict that theory. If the explicit object representation is distinct from the features themselves, wouldn't it follow that the unimodal features should remain unchanged? Explaining how the results help discriminate between the two representational theories raised in the introduction, and how it specifically supports the explicit integrative theory, would enable the reader to contextualize the reported findings.

We hope that our changes throughout the manuscript now better clarify that we are looking for a crossmodal integrative code different from the unimodal features. We remain open to alternative structures of the integrative code, and so we have not predicted a direction of representational change (though we provide some interpretation in the Discussion section). As mentioned in a previous response, these changes are described on pages 12 and 13.

Notably, it is likely the case that the crossmodal integrative code can change *in addition to* the unimodal feature representations. For example, we have previously shown that unimodal visual feature representations are influenced by experience in parallel to the representation of the conjunction (e.g., Liang et al., 2020; *Cerebral Cortex*).

– The finding that unimodal features correlate with their respective multimodal object representations before but not after learning in the perirhinal cortex provides the best support for the claim that object representations are independent of feature representations. However, there are two modifications to the current analysis that could make this argument more direct. First, readers would want to know that unimodal features no longer correlate with congruent objects in the perirhinal cortex, but that their correlations with incongruent objects are unaffected. Otherwise, we can't interpret this result as due to multimodal object learning per se. Second, there's the question of representational stability in the perirhinal cortex. If perirhinal feature representations are not stable across days, it is possible that featural content is actually present in the object representations, but this would only be evident if one used the Day 4 feature representations instead of the Day 2 feature representations. If the Day 4 feature representations do not correlate with the Day 4 congruent object representations, this would be the most direct evidence for explicit, integrative object representations that are distinct from feature representations in the perirhinal cortex.

We thank the reviewer for this suggestion and have conducted the suggested analyses, splitting the congruent and incongruent conditions. There was no significant interaction between modality and congruency in any ROI across days or within days. In one possibility, it may be the case that both congruent and incongruent crossmodal objects are represented differently from their underlying unimodal features, and all of these representations can change with experience (e.g., are not stable across days).

Indeed, in an additional exploratory analysis, we found that perirhinal cortex was the only region where the Day 4 unimodal feature representations do not correlate with the Day 4 crossmodal object representations – suggestive of a crossmodal integrative code transformed from the unimodal features. Finally, we emphasize that unimodal feature representations can also change with learning in parallel to changes at the level of the conjunction (e.g., Liang et al., 2020; Cerebral Cortex).

“To examine whether these results differed by congruency (i.e., whether any modality-specific biases differed as a function of whether the object was congruent or incongruent), we conducted exploratory linear mixed models for each of the five a priori ROIs across learning days. More specifically, we correlated: (1) the voxel-wise activity for Unimodal Feature Runs before crossmodal learning to the voxel-wise activity for Crossmodal Object Runs before crossmodal learning (Day 2 vs. Day 2), (2) the voxel-wise activity for Unimodal Feature Runs before crossmodal learning to the voxel-wise activity for Crossmodal Object Runs after crossmodal learning (Day 2 vs Day 4), and (3) the voxel-wise activity for Unimodal Feature Runs after crossmodal learning to the voxel-wise activity for Crossmodal Object Runs after crossmodal learning (Day 4 vs Day 4). For each of the three analyses described, we then conducted separate linear mixed models which included modality (visual feature match to crossmodal object vs. sound feature match to crossmodal object) and congruency (congruent vs. incongruent).” – pg. 10

“There was no significant relationship between modality and congruency in any ROI between Day 2 and Day 2 (F_1,346-368_ between 0.00 and 1.06, p between 0.30 and 0.99), between Day 2 and Day 4 (F_1,346-368_ between 0.021 and 0.91, p between 0.34 and 0.89), or between Day 4 and Day 4 (F_1,346-368_ between 0.01 and 3.05, p between 0.082 and 0.93). However, exploratory analyses revealed that perirhinal cortex was the only region without a modality-specific bias and where the unimodal feature runs were not significantly correlated to the crossmodal object runs after crossmodal learning (Figure 5 —figure supplement 2).” – pg. 11

“Taken together, the overall pattern of results suggests that representations of the crossmodal objects in perirhinal cortex were heavily influenced by their consistent visual features before crossmodal learning. However, the crossmodal object representations were no longer influenced by the component visual features after crossmodal learning (Figure 5, Figure 5 —figure supplement 2). Additional exploratory analyses did not find evidence of experience-dependent changes in the hippocampus or inferior parietal lobes (Figure 5 —figure supplement 1).” – pg. 11

“The voxel-wise matrix for Unimodal Feature runs on Day 4 were correlated to the voxel-wise matrix for Crossmodal Object runs on Day 4 (see Figure 5 in the main text for an example). We compared the average pattern similarity (z-transformed Pearson correlation) between shape (blue) and sound (orange) features specifically after crossmodal learning. Consistent with Figure 5b, perirhinal cortex was the only region without a modality-specific bias. Furthermore, perirhinal cortex was the only region where the representations of both the visual and sound features were not significantly correlated to the crossmodal objects. By contrast, every other region maintained a modality-specific bias for either the visual or sound features. These results suggest that perirhinal cortex representations were transformed with experience, such that the initial visual shape representations (Figure 5a) were no longer grounded in a single modality after crossmodal learning. Furthermore, these results suggest that crossmodal learning formed an integrative code different from the unimodal features in perirhinal cortex, as the visual and sound features were not significantly correlated with the crossmodal objects. * p < 0.05, ** p < 0.01, *** p < 0.001. Horizontal lines within brain regions indicate a significant main effect of modality. Vertical asterisks denote pattern similarity comparisons relative to 0.” – Figure 5 —figure supplement 2

– There is a conflation between "visual features" and "objects" throughout the manuscript which can be quite confusing. Sometimes the word "object" is used to represent multimodal visual-auditory objects ("multimodal object"), other times "object" refers to complex visual objects with multiple features ("frog"), and sometimes simple visual stimuli (in functional localizer). In particular, the frog example used throughout the manuscript doesn't feel appropriate, because the representation of "frog" is a lot more complex than one visual feature, and is already an integrated representation across different visual features and modalities (even if the "croak" feature is hypothetically removed). The frog example also muddies the theoretical claims-the authors want the "frog" representation to change when paired with "croak" because "frog" is already an integrated object representation that now needs to be modified. However, the authors should *not* want the representation of a single visual feature to change, since one visual feature is an ingredient that is fed into a separate, integrated object representation.

We thank the reviewer for providing this suggestion and we now replace “features” with unimodal features and “object” with crossmodal object throughout the manuscript text, figures, and title where appropriate (which was also suggested by Reviewer 3). We also removed the “frog” example in the results.

Reviewer #3 (Recommendations for the authors):I would encourage the authors to provide open access to the data and analysis code.

We thank the reviewer for the positive comments on our work. The preprocessed data files for the univariate and pattern similarity analyses are available on OSF. Given the non-standard format of the multi-echo pipeline and large storage requirements needed, we hope to make the data files readable and openly available in the future when file standards have evolved to include multi-echo ICA data (e.g., BIDS).